# Putting people in context: ERP responses to bodies in natural scenes

**Ilya Nudnou, Abigail Post, Alyson Saville, Benjamin Balas**[ID]¤*

Department of Psychology, Center for Visual and Cognitive Neuroscience, North Dakota State University, Fargo, ND, United States of America

¤ Current address: Department of Psychology, North Dakota State University, Fargo, ND, United States of America
* Benjamin.balas@ndsu.edu

## Abstract

The N190 is a body-sensitive ERP component that responds to images of human bodies in different poses. In natural settings, bodies vary in posture and appear within complex, cluttered environments, frequently with other people. In many studies, however, such variability is absent. How does the N190 response change when observers see images that incorporate these sources of variability? In two experiments (N = 16 each), we varied the natural appearance of upright and inverted bodies to examine how the N190 amplitude, latency, and the Body-Inversion Effect (BIE) were affected by natural variability. In Experiment 1, we varied the number of people present in upright and inverted naturalistic scenes such that only one body, a subitizable number of bodies, or a "crowd" was present. In Experiment 2, we varied the natural body appearance by presenting bodies either as silhouettes or with photographic detail. Further, we varied the natural background appearance by either removing it or presenting individual bodies within a rich environment. Using component-based analyses of the N190, we found that the number of bodies in a scene reduced the N190 amplitude, but didn't affect the BIE (Experiment 1). Naturalistic body and background appearance (Experiment 2) also affected the N190, such that component amplitude was dramatically reduced by naturalistic appearance. To complement this analysis, we examined the contribution of spatiotemporal features (i.e., electrode × time point amplitude) via SVM decoding. This technique allows us to examine which timepoints across the entire waveform contribute the most to successful decoding of body orientation in each condition. This analysis revealed that later timepoints (after 300ms) contribute most to successful orientation decoding. These results demonstrate that natural appearance variability affects body processing at the N190 and that later ERP components may make important contributions to body processing in natural scenes.

## Introduction

People are very good at detecting people in the world around them. Both faces and bodies are detected quickly and accurately in a range of different tasks. Faces, for example, "pop out" in

**Data Availability Statement:** The data are available on the Open Science Framework (https://osf.io/4su8x/).

**Funding:** The author(s) received no specific funding for this work.

**Competing interests:** The authors have declared that no competing interests exist.

visual search tasks [1] and this effect is robust to flanking elements and presentation of faces in the visual periphery [2]. Bodies also appear to be privileged in some sense when it comes to various detection tasks. In a standard inattentional blindness paradigm, body images enjoy an advantage compared to control object categories [3] and bodies also appear to engage attention more readily in speeded categorization tasks [4]. The human visual system thus appears to treat the visual components of people, faces and bodies, with special care: People are detected rapidly, attention is deployed to people quickly (perhaps obligatorily), and category information about faces and bodies is recovered accurately.

Additional behavioral and neural results support the hypothesis that faces and bodies enjoy a privileged status in visual processing that distinguishes them from other object categories. Behaviorally, both face and body perception are dramatically affected by image inversion [5–8] while other object categories are generally more robust to orientation changes. Faces and bodies are also both processed holistically [9,10], which broadly refers to a processing strategy that relies predominantly on large-scale templates of object appearance. This is evident for both faces and bodies in tasks that use versions of the composite face effect [11] or part-whole tasks [12], both of which are established paradigms that make it possible to measure advantages for processing of an entire object relative to processing of object parts. These behavioral indications that face and body perception depend on unique mechanisms are further supported by a substantial amount of evidence that both faces and bodies are represented in extrastriate visual cortex by specialized cortical areas [13]: For faces, the fusiform face area (or FFA) and other loci within an extended face network, and for bodies, the extrastriate body area or fusiform body area (EBA–[14,15]. Face and body perception are also associated with distinct electrophysiological responses that are measurable via EEG/ERP. Faces reliably elicit an ERP component referred to as the N170 [16], for example, while bodies elicit a similar component referred to as the N190 [17]. Both of these ERP components exhibit clear inversion effects [18,19], linking the behavioral results described above to specific neural mechanisms. Further, evidence from intracranial recordings [20] and patterns of response to stimuli depicting different body parts [21] indicate good correspondence between the body-selective ERP responses and activity in the extrastriate cortical areas selective for body stimuli [20].

An important caveat about nearly all of this work, however, is that person perception is rarely examined in the context of real-world environments. Specifically, prior studies have primarily focused on the perception of images that include a single object, presented in isolation or in a structured array (as in search tasks), often with variables such as viewing angle, lighting, and other sources of image variation eliminated, tightly controlled, or explicitly parameterized because particular sources of variability may be of specific interest. Person perception under these controlled circumstances may be rather different than how it plays out "in the wild" where faces and bodies appear in wide-ranging poses and complex environments. Besides a generic concern about ecological validity, there are a number of more specific reasons why studying person perception and face/body categorization using simplified tasks and stimuli like this may not be representative of the way people are perceived in real-world visual environments. For example, the presence of clutter in the form of dense backgrounds and/or the presence of non-target objects can dramatically impact observers' abilities to recognize target objects, including faces and bodies. In particular, visual crowding, which refers to the phenomenology of flanking items making the recognition of a peripherally-viewed target more difficult, operates both at low- and high-level stages of visual processing. Faces, for example, suffer from 'holistic crowding' that reflects the interaction of high-level representations of flanker and target faces [22,23], demonstrating that at the level of object processing, flanking elements can dramatically impair recognition performance. More generally, complex objects suffer from the effects of visual crowding in much the same way that simpler objects like letters and

oriented bars do [24]. However, clutter can also provide context for object categorization. People do not appear at random positions in natural scenes, after all, so the structure of a natural scene can provide statistical cues regarding the likely location of faces and bodies in an image [25], which can guide the deployment of attention or eye movements to the people in a scene [26]. Indeed, observers tend to fixate scenes centrally first, then quickly make additional eye movements to faces and bodies (even those that lack faces) in natural scenes [27]. The presence of bodies in natural scenes can be distinguished from the presence of cars using decoding analyses of MEG data [28] which further indicates that despite the potential masking or crowding effects of a cluttered scene, visual processing of bodies can nonetheless proceed rapidly. Clutter may sometimes be context, then, and likewise natural variability in body appearance may be a boon or a detriment. Ultimately, the fact that body appearance in natural environments is subject to a broad range of factors like this that we frequently remove from our stimuli means that person perception may have been largely characterized by systematically removing many of the conditions that they must function under nearly all the time.

With regard to the neural mechanisms supporting person perception, these issues are of particular importance. By limiting the variability of person images (faces or bodies) that we present to observers in tasks designed to examine the response properties of neural markers of person perception, we not only limit the generalizability of our results to real environments, but run the risk of mis-characterizing the neural basis of person perception. This risk was made particularly apparent several years ago when it was proposed that the face-specificity of the N170 ERP component could be artifactual [29]. Specifically, while one account of the larger amplitude of this component to images of faces was based on the sensitivity of this response to faces as a distinct object category, the authors introduced a competing account based on differences in the variability of stimuli within target and non-target categories. That is, the N170 may not reflect the responses of a face-specific neural locus, but rather the effect of averaging responses to images that vary little in appearance across trials, leading to reduced variability in the underlying neural response. On balance, this alternative account appears to have been countered by a number of subsequent studies supporting the role of N170 in face processing proper [30,31], but nonetheless there is an important point made here regarding the real appearance of objects: Ignoring variability and its role in shaping neural responses may lead to mischaracterizations of neural selectivity.

Our goal in the current study is to examine the electrophysiological response to images containing people with an eye towards examining the effects of wider-ranging natural variability in real-world images. We do this in two ERP experiments, each designed to reveal how the N190 component is affected by variation in body appearance in natural scenes. In each case, we use images with sufficient natural variability that we are not able to isolate the effects of our manipulations of body and scene appearance to body-selective processes specifically, but rather ask how these manipulations affect the N190. However, in both cases we also use body inversion as a means of controlling for low-level appearance and measuring the neural body-inversion effect and its potential interactions with our other stimulus variables.

In our first experiment, we hypothesized that the N190 may exhibit a dose-dependent response such that more bodies in a scene may lead to a larger component amplitude and that consistent inversion effects should be observed across images with varying numbers of people in them. In our second experiment, we removed foreground and background detail from our images to examine how the naturalistic appearance of bodies and of background scenes might affect the N190 response. Here, we hypothesized that natural detail might enhance the N190 response and lead to a larger inversion effect due to naturalistic levels of detail more closely matching real experience with bodies. In both cases, the interactions between stimulus orientation (upright vs. inverted orientation) and our appearance variables (numerosity in

Experiment 1, scene/body texture in Experiment 2) are of critical interest. While the impacts of low-level changes in scene and object appearance measured at this component are potentially interesting in their own right, these interactions support potential accounts of the data that are specific to body processing rather than a by-product of earlier visual processing. We also complement our analysis of the N190 component in each task with neural decoding analyses, which allow us to examine the discriminability of ERP responses to our different stimulus categories without the need to isolate a specific component of the ERP response spatially or temporally.

## Experiment 1

In our first experiment, we examined how the N190 was affected both by inversion and by varying the number of bodies in a scene. In most cases, the use of natural images in studies of neural face or body processing still tends to involve images in which the face of a single person takes up nearly the entire image [32–34], though see [35] for a nice example of cluttered stimuli containing people). This isn't particularly representative of visual experience, where observers frequently encounter people in groups that are at varying eccentricities and distances, and limits the extent to which person perception is really being studied "In the Wild" where noisy backgrounds and multiple objects roam. With regard to body perception, several results demonstrate that the appearance of multiple bodies affects perceptual processing in systematic ways. Bodies that are facing each other are subject to a classic inversion effect (poorer categorization for inverted images) that is not obtained for images depicting bodies that face away from one another, for example [36]. This effect has been replicated behaviorally and also observed via fMRI responses to body dyads in lateral occipital cortex [37], which also exhibits differential responses to single bodies vs. dyads in different sub-regions. This facing-bodies inversion effect is specifically related to body perception, as evidenced by the persistence of the key effect (larger inversion effects for facing dyads) when there is no face information included [38]. Though these results specifically concern the perception of bodies engaged in social interactions (a feature of natural images with bodies we do not examine here), the pattern of results is clearly robust and suggests that body numerosity can affect behavioral and neural processing of body images.

### Methods

**Participants.** We recruited 16 participants (7 female) to take part in this experiment. This sample size was deemed appropriate based on an apriori power analysis carried out using G*Power based on effect sizes estimated from Thierry et al. [17] and Borhani et al. [39]. This analysis indicated that a sample size of 14 would provide us with 90% power to detect effect sizes of moderate size, consistent with these previous reports. All participants were recruited from the STEM cohort of the North Dakota State Governor's School during the summer of 2019 and were between the ages of 16–19 years old. Though this makes our sample somewhat younger than the more typical undergraduate or young adult sample, the relatively small difference in age between our participants and this commonly used population suggests to us that there are unlikely to be substantial differences in behavioral or neural processing that would limit the generalizability of our conclusions. Written informed consent was obtained from participants' parents and written assent was obtained from the participants themselves, save for participants who were 18 years or older and were thus able to provide written informed consent themselves. All participants self-reported normal or corrected-to-normal visual acuity and no history of neurological impairment. We assessed handedness using the Edinburgh Handedness Inventory [40] and found that no participants were scored as strongly left-

handed. All procedures for recruitment and testing were approved by the NDSU IRB and written informed consent was obtained from all participants.

**Stimuli.** We selected images from the Penn-Fudan pedestrian database [41] for use in Experiment 1. This database contains a large variety of images depicting pedestrians in two different cities with body position, pose, size, and numerosity varying widely across images. Also, the metadata accompanying the database includes segmentation masks for each pedestrian in every image, making it possible to easily remove the background of each image and/or present bodies as silhouettes or in full photographic detail. For the purposes of this first experiment, we chose to remove the background of each image we selected and render pedestrians as silhouettes rather than with full natural detail. To vary body numerosity, we selected images containing (1) A single pedestrian, (2) a small, subitizable number of pedestrians (<3), and (3) A crowd of people, too many in number to be subitized (>5). The original images varied in size, but were cropped and resized to a common size of 512 x 512 pixels. We selected a total of 32 images per numerosity category. We did not control for the visibility of individual faces in images. As we do not have permission to share the stimulus images themselves, we encourage readers to visit the database homepage (https://www.cis.upenn.edu/~jshi/ped_html/) to see the original images. In Fig 1, we display an image similar to those in our Many People condition in both Upright and Inverted orientation.

**Procedure.** We asked participants to complete a simple orientation categorization task using the pedestrian images described above. During EEG recording, we presented upright and inverted (vertically flipped) versions of these images in a pseudo-randomized order and asked participants to indicate whether the image was upright or upside-down via a button box that they held in their lap during the testing session. Each image was presented for 500ms, followed by a fixation cross that remained visible until participants made their response. Next, we imposed an interstimulus interval of random duration that was from a uniform distribution bounded between 800-1500ms. Participants were seated approximately 57 cm away from a 1024 x 768 LCD monitor during the task and the stimuli subtended approximately 10 degrees at this distance. Upright and inverted versions of each of our stimuli were presented during the session for a grand total of 64 trials per condition and 384 trials in the entire experiment. All stimulus presentation and response collection routines were implemented in EPrime v2.0.

### Upright (Many People image)          Inverted (Many People image)

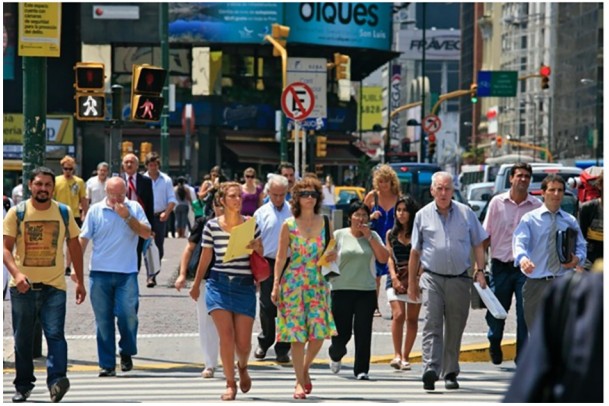 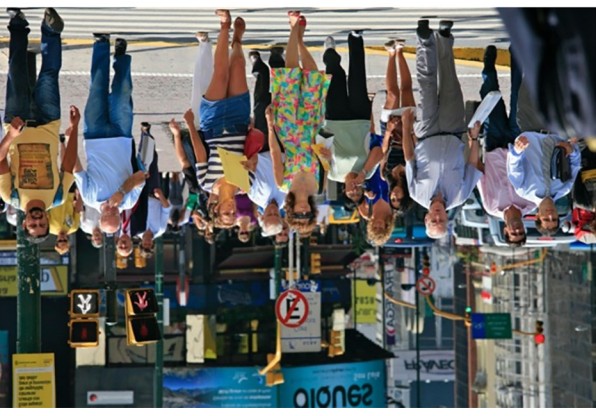

**Fig 1. Representative images illustrating the inversion manipulation in Experiment 1.** Stimulus images also varied in the number of people present in the photo–Single Person images depicted only one pedestrian, Few People images depicted 2 or 3 individuals, and Many People images depicted 4 or more. Inversion was achieved by vertically flipping each image. Image Credit: By Alex Proimos from Sydney, Australia—CrossWalk, CC BY 2.0, https://commons.wikimedia.org/w/index.php?curid=25650842.

We recorded EEG using 64-Channel Hydrocel Geodesic Sensor Nets manufactured by Electrical Geodesics (EGI) and an EGI 400 NetAmps amplifier. An appropriately sized net was selected for each participant according to their head circumference and the net was placed on each participant by marking the position of their scalp vertex with a grease pencil and aligning the net's vertex electrode with this mark. Prior to net application, sensor nets were soaked in a warm potassium chloride solution for 5 minutes. After application, we assessed the impedance at each electrode and prepped individual electrodes with additional KCl solution and/or scrubbing against the scalp until we achieved impedances below 25 kΩ across the entire net.

During EEG recording, participants were seated in an electrically shielded room to minimize contamination of the EEG data by external sources of electrical noise. We sampled continuous EEG activity at 250 Hz and referenced activity to the vertex electrode. The entire testing session for Experiment 1 lasted approximately 25 minutes.

**EEG preprocessing.**    To obtain ERPs from each of our participants we carried out a series of pre-processing steps on the continuous EEG data using NetStation v5.0. First, the raw data was bandpass filtered with cut-off frequencies at 0.1Hz and 30Hz. Subsequently, we segmented the continuous EEG data into individual trials using inline triggers inserted into the EEG record to mark stimulus onset. Each segment began 100ms before stimulus onset to obtain a pre-stimulus baseline and ended 1000ms after stimulus onset for a total of 1100ms per epoch. Each segment was then baseline corrected by calculating the average baseline voltage in the 100ms pre-stimulus onset period and subtracting this value from the entire segment. Next, we carried out artifact detection and removal using NetStation's default settings for eye blink, eye movement, and bad channel detection. After the application of these algorithms, we replaced any bad channels using spherical spline interpolation (again, using NetStation's default settings for this procedure) and calculated average ERPs for each subject in each of the experimental conditions. Finally, we re-referenced these data to an average reference, yielding the final ERPs per subject.

To measure the latency and amplitude of the N190, we identified sensors of interest and a time interval of interest using the grand average ERP calculated across all participants and collapsed across experimental conditions. Visual inspection of the scalp distribution led us to select 6 electrodes (3 per hemisphere) over occipito-temporal scalp regions where we observed the largest negative deflection consistent with N190 morphology: In the left hemisphere, we selected electrodes 29, 32 and 33, while in the right hemisphere we selected electrodes 43, 44, and 47. These numbers refer to the indices used in EGI's Hydrocel Geodesic Sensor Nets and full correspondence between these numbers and the 10–20 system can be found in [42]. Briefly, however, these electrodes correspond to P7 and P8 and two neighboring sites, which is consistent with the initial report of the N190 by Thierry et al. (2006)At these electrodes, we determined that a time interval of 140ms - 216ms post-stimulus onset was sufficient for capturing the N190 component across conditions and participants by inspection of the grand average ERP at these sites (Fig 2). This was determined by inspecting the grand average waveform across the sensors of interest and selecting timepoints at which voltage was halfway between the preceding (or subsequent) voltage peak and the negative peak of our putative N190 component. While this time interval is shifted slightly earlier than Thierry et al's. [17] report of a body-sensitive component peaking at N190, this measurement window was selected because it adequately captured the rise and fall of the component we measured at the sensors of interest.

## Results

**Behavioral performance.**    Across all conditions, accuracy at reporting the orientation of the stimuli (upright or inverted) exceeded 96% with response latencies to correct responses

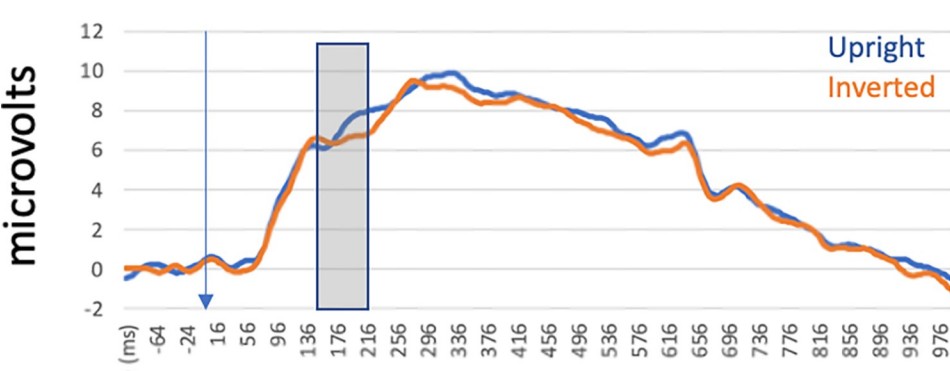

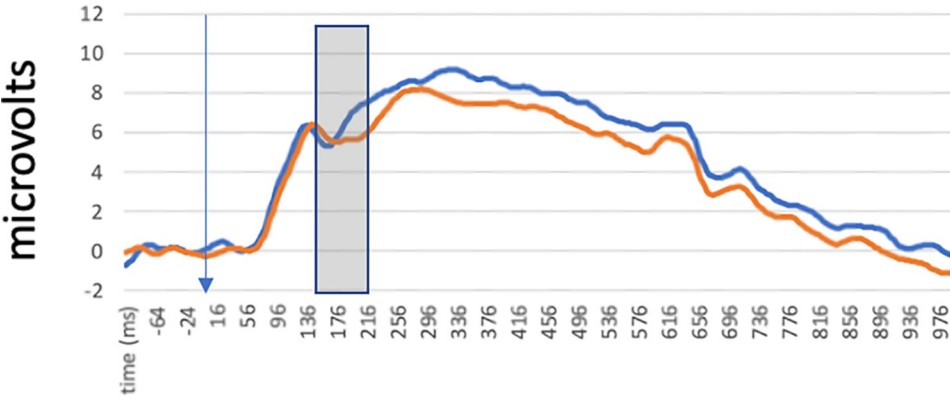

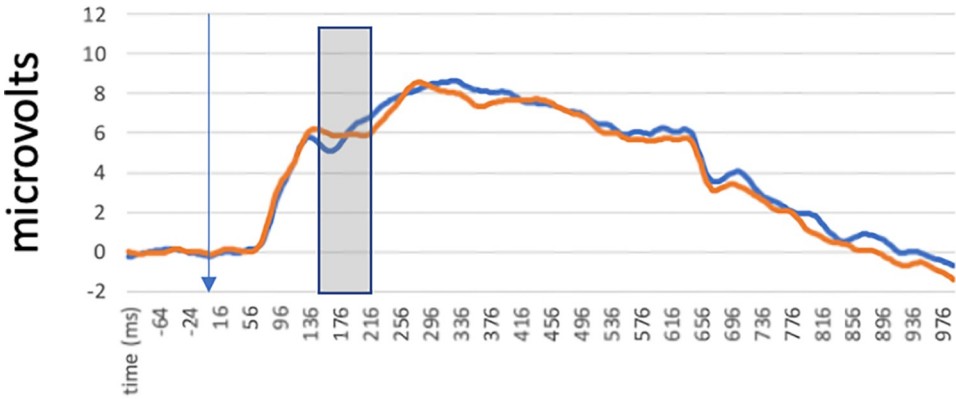

**Fig 2. Grand average ERPs calculated across participants in all conditions in Experiment 1.** These data arecollapsed across left and right hemisphere sensors to highlight the effects of image orientation and body numerosity. The transparent grey window indicates the approximate time window used to analyze the N190 component.

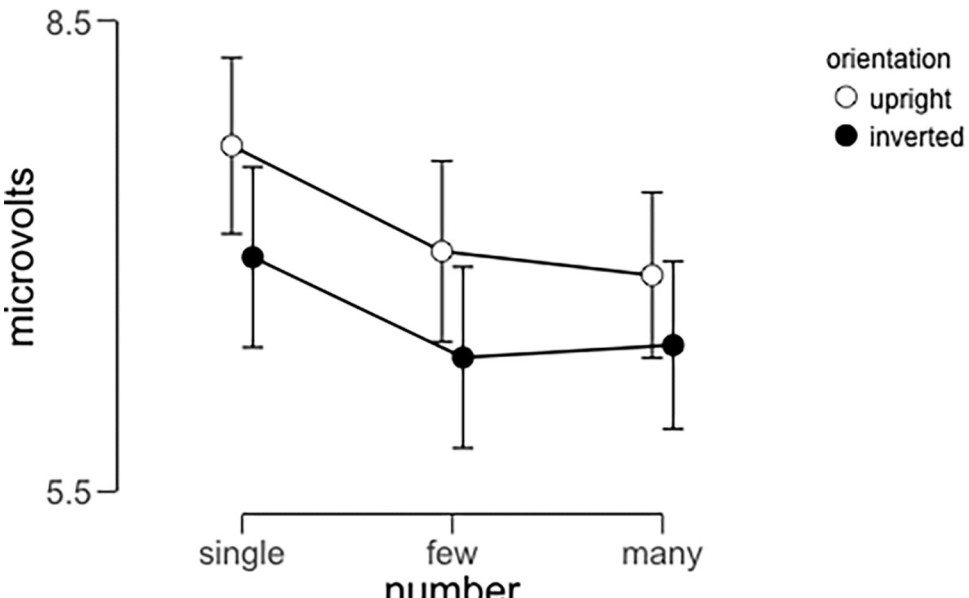

**Fig 3. Average N190 amplitude across conditions and participants in Experiment 1.** Error bars represent 95% confidence intervals.

under 600ms for all conditions. Given that this task was clearly extremely easy for our participants and completed with high accuracy, we do not include further discussion of this data.

**N190 amplitude.** Within the time interval described above, we calculated the mean amplitude for all conditions for each of our participants and analyzed these values using a 3x2x2 repeated-measures ANOVA with body numerosity (single, few, many), orientation (upright vs. inverted) and hemisphere (left vs. right) as within-subjects factors. We implemented this test using JASP [43] This analysis revealed main effects of numerosity ($F(1,15) = 7.64$, $p = 0.014$, $\eta^2_p = 0.34$) and orientation ($F(1,15) = 7.19$, $p = 0.003$, $\eta^2_p = 0.32$). The main effect of numerosity was the result of significant differences between N190 amplitude in the "single" condition compared to the "few" condition ($t = 4.46$, Cohen's $d = 1.12$, $p_{bonf} < 0.001$) and also in the "single" condition compared to the "many" condition ($t = 4.21$, Cohen's $d = 1.05$, $p_{bonf} < 0.001$). The difference between amplitude in the "few" condition compared to the "many" condition did not reach significance. The main effect of orientation was the result of larger positive amplitudes in the upright condition compared to the inverted condition ($t = 4.46$, Cohen's $d = 1.11$, $p_{bonf} < 0.001$). These main effects are illustrated in Fig 3. We also observed a significant interaction between orientation and hemisphere ($F(1,15) = 7.52$, $p < 0.001$, $\eta^2_p = 0.54$). Pairwise post-hoc tests (corrected for a family of 6 comparisons) revealed that the only significant difference between combinations of orientation and hemisphere resulted from the comparison between upright and inverted values in the right hemisphere ($t = 4.192$, Cohen's $d = 0.26$, $p = 0.003$). The critical interaction between orientation and number did not reach significance ($F(2,30) = 0.472$, $p = 0.68$).

**N190 peak latency.** Within the time interval described above, we described N190 latency to peak by identifying the most negative value within the critical window per participant. We analyzed these values using a 3x2x2 repeated-measures ANOVA with the same structure as described above for our analysis of the N190 mean amplitude. This analysis revealed only a main effect of orientation ($F(1,15) = 20.55$, $p < 0.001$, $\eta^2_p = 0.58$) driven by faster latencies in response to upright bodies compared to inverted bodies ($t = 5.53$, $p_{bonf} < 0.001$, Cohen's

d = 1.38). Again, the critical interaction between orientation and number did not reach significance (F(2,30) = 1.91, p = 0.17).

**Decoding analysis.** As well as grand average ERP analyses, we classified the average spatiotemporal maps (i.e., electrode × time point amplitude) of each condition in a pairwise manner. The contribution of spatiotemporal features to decoding was estimated with the aid of support vector machine (SVM) weights. This method for feature selection is instrumental in the effort to decrease the dimensionality of relevant observations, to boost discriminability and to estimate feature diagnosticity [44,45]. The SVM procedure is able to linearly separate classes with a hyperplane within a high-dimensional representation of their data points. In the current study, we linearly separated each condition from another condition in a pairwise manner. The most diagnostic points, which are the closest to the hyperplane, are assigned the highest weights. Each such decoding analysis was performed in 63 steps (half the number of bins, with 1 bin = 4ms), with the most diagnostic features removed during each step so as to examine finer-grained contributions than are possible after just one run. This process results in a ranking assignment, with the most diagnostic features ranked the most highly (Fig 4). After the procedure is finished, we are able to average the rank of each feature, and reshape the feature vector into two dimensions for visualization purposes (Fig 5). This is different from normal SVM, where generally the weight values after a single run are visualized, though the resulting spatiotemporal maps often look similar due to important features consistently being ranked highly. To avoid overfitting, we used the leave-one-trial-out method of cross validation (train on 63, test on 1). While we did not exhaustively iterate through all such trial combinations, we performed 150 iterations of randomly selecting all but one trial for training and the remaining trial for testing from each pairwise combination of conditions. Eye and frontal electrodes (5, 10, 62, 63) were not used for this analysis, due to the consistently high ranking achieved by the features from these channels, which are unlikely to reflect visual perceptual contribution. The Matlab function fitcsvm was used with default parameters and the linear kernel.

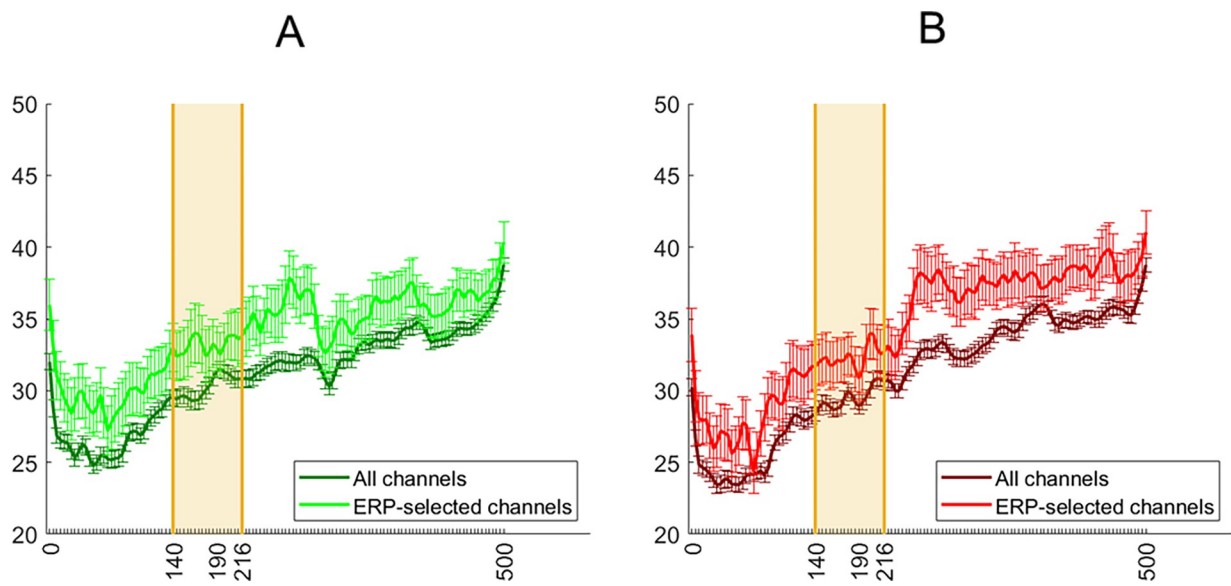

**Fig 4.** Ranking of temporal features, averaged across electrodes for decoding numerosity between upright conditions (A), and decoding orientation in matching numerosity conditions (B). Orange shading represents the chosen interval around N190. Again, while there are moderately highly-ranked features in the same spatial and temporal neighborhood as the N190, there are also multiple additional features that were also ranked highly.

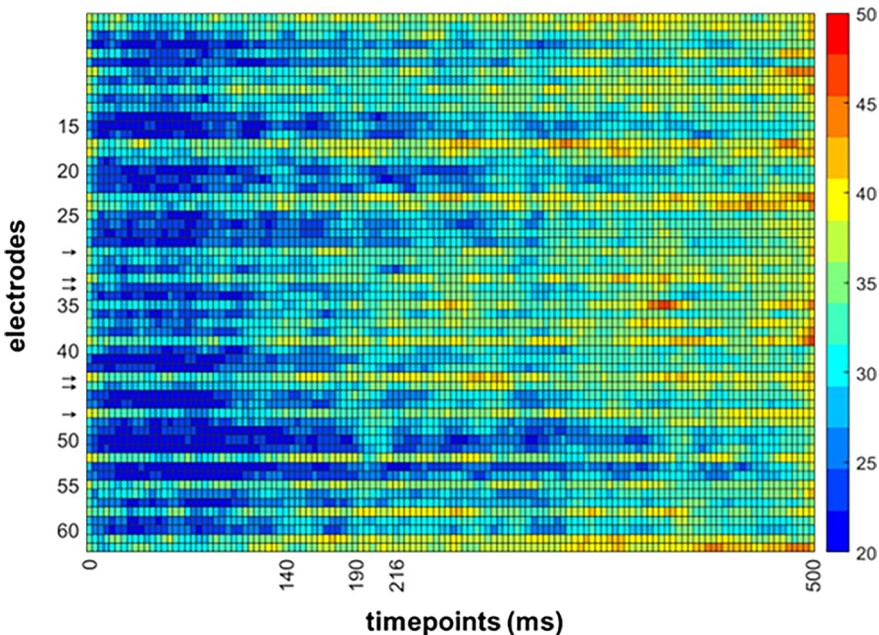

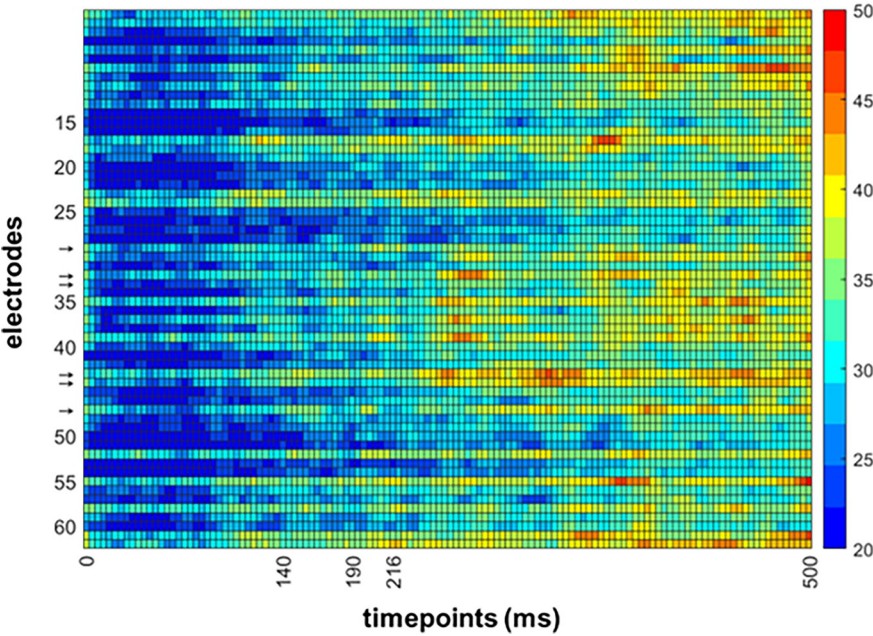

**Fig 5. a.** Average ranking of spatiotemporal features (electrode x timepoint) for decoding body numerosity across upright conditions. Inverted conditions were not included in this analysis. Warmer colors represent spatiotemporal bins that were ranked higher on average across all iterations of our classification analysis. While there are moderately highly-ranked features in the same spatial and temporal neighborhood as the N190, there are also multiple additional features that were also ranked highly. → = Electrodes used in the ERP analysis. **b.** Average ranking of spatiotemporal features (electrode x timepoint) for decoding orientation in matching numerosity conditions (Upright Single vs. Upright Few, e.g.) and collapsed across all numerosities. Warmer colors represent spatiotemporal bins that were ranked higher on average across all iterations of our classification analysis. Again, while there are moderately highly-ranked features in the same spatial and temporal neighborhood as the N190, there are also multiple additional features that were also ranked highly. → = Electrodes used in the ERP analysis.

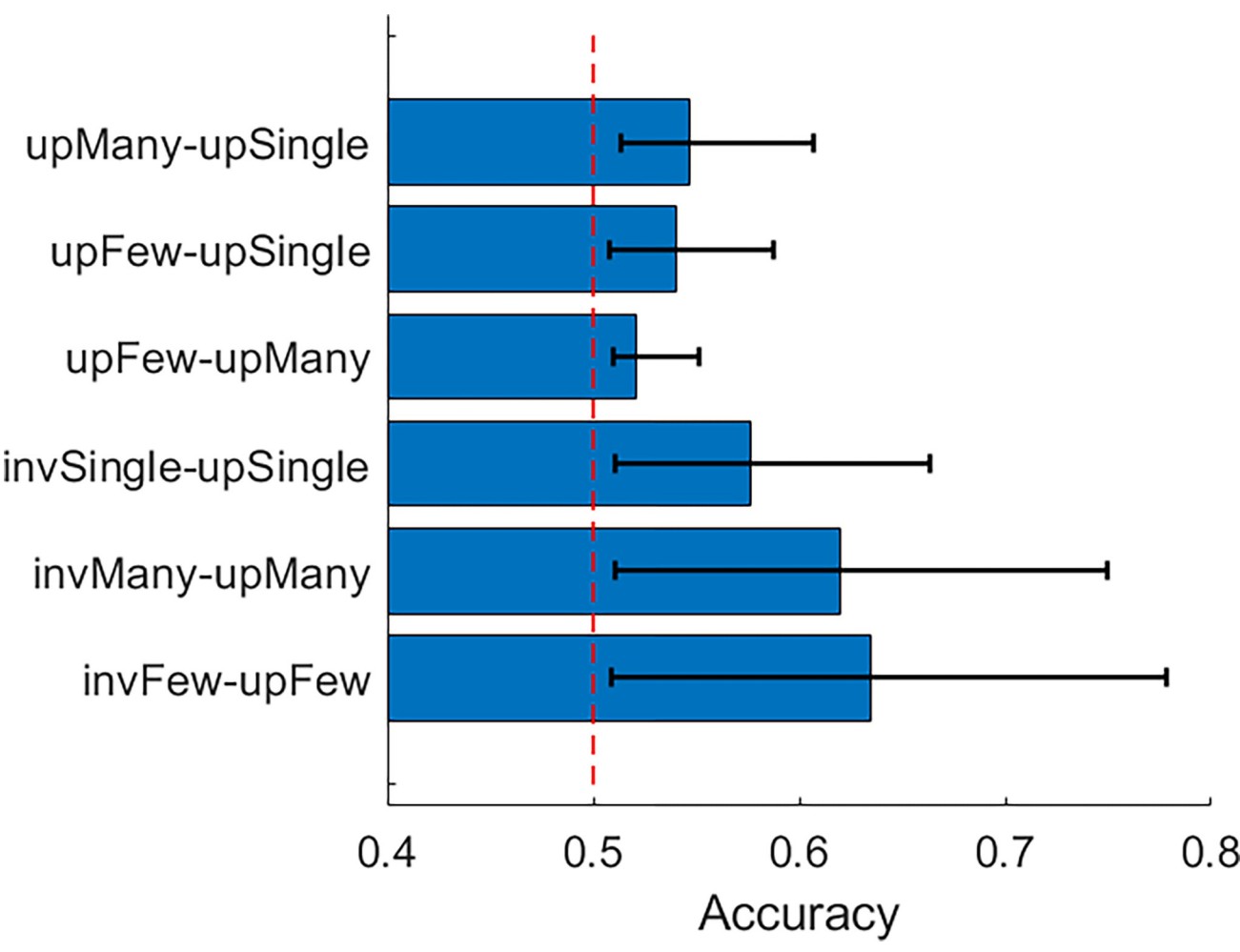

**Fig 6. Pairwise decoding accuracy for all comparisons in Experiment 1.** Up: Upright; Inv: Inverted; Single: One body; Few: A few bodies; Many: Multiple bodies. Error bars represent 95% CIs.

In order for the feature contributions to be interpretable, the decoding accuracy must actually be above chance, which in a pairwise comparison equals 50%. This was achieved for all condition comparisons (Fig 6). Furthermore, there is a clear increase in accuracy when classifying orientation instead of body numerosity. Overall, the decoding results reveal the contribution of features outside the time interval for the N190.

## Discussion

Our first experiment replicated previously observed properties of the body-sensitive N190 component and also revealed novel sensitivity to body numerosity. We were able to observe a consistent body inversion effect across all conditions, with the amplitude of the N190 significantly reduced, delayed, and potentially diffused by inverted images of bodies. The effect we observed with regard to amplitude is intriguing in that there are reports of both a classic inversion effect for body images in which the amplitude of the N190 is increased (becomes more negative) subject to body inversion (see [19,46,47] and an "inverted" inversion effect in which the amplitude is reduced [19]. This latter outcome has been linked to the absence of heads and/or eyes in body images [48], which may indicate that the enhancement of the N190

amplitude with inversion depends critically on a canonical configuration of the human body and face [49]. With regard to the stimuli used in Experiment 1, the high variability in body posture, including rotations in depth such that the head and face may have been facing away from the camera, may have been sufficient to disrupt this canonical body configuration. If this is the case, our results are an important indication that it is not only dramatic disruption of the body schema via the selective removal of body parts that affects the N190 body inversion effect, but more prosaic and ecologically valid variability in body appearance resulting from simple postural changes during locomotion.

Another feature of our data that differs from previous reports is the overall amplitude of the N190 component. While we did observe a clear negative deflection in the same approximate time interval as previous reports, the amplitude of the N190 across all conditions in our study was smaller than expected. While the absolute amplitude of ERP components is not a useful index of the neural response [50] the fact that we observed only a small negative deflection rather than a large enough response to lead to negative average amplitudes is an indicator that the N190 response we observed was small compared to previous literature. We will be able to comment on this more following the presentation of the results of Experiment 2, but this is an indicator that the presence of a complex, natural background may also have important consequences for the magnitude and timing of the N190 component.

Finally, Experiment 1 also revealed that body numerosity has a clear effect on N190 amplitudes, with more negative responses to images containing more bodies. This enhancement of the N190 amplitude could suggest a dose-dependent response in which each body makes an independent contribution to the aggregate neural response. Alternatively, it could also be the case that the presence of more bodies in a complex scene makes it more likely that there will be at least one individual who appears in a more canonical pose, perhaps with eyes and face visible, leading to a larger N190. We cannot distinguish between these possibilities at present, but instead emphasize that the response of the N190 does depend on the presence of multiple people in a scene. Our results are consistent to some extent with Bellot, Abassi & Papeo's [51] observations of enhanced responses in the extrastriate body area for facing bodies, though we note that our images varied much more widely in terms of the facing interactions depicted in the scene stimuli. Despite this, an enhanced N190 amplitude to multiple bodies is consistent with the proposal that dyads or multiple groups of bodies may have a unique status in body perception [52]. The fact that this effect is an enhancement of the N190 amplitude is at odds with what one might predict based on the effects of inter-stimulus perceptual variability (ISPV) on ERP component amplitude [29]. Specifically, if body configuration and pose affect the N190 amplitude, the presence of multiple bodies with different postures could lead to a more diffuse N190 response with an associated lower amplitude in the aggregate based on temporal jitter of the responses to individual bodies. This is different than Thierry et al.'s original discussion of ISPV across multiple trials in that we are considering the presence of multiple bodies within a single image as another potential case of jittered individual response being summed to produce a single ERP component, but the analogy is a useful way to rule out one simple story about how multiple bodies in an image could produce changes in the N190 amplitude.

## Experiment 2

In Experiment 2, we examined how the natural appearance of bodies in scenes affected the N190 ERP component. We varied both the natural appearance of pedestrians and the presence of a complex natural background to determine how scene context affected ERP responses to bodies. We hypothesized that providing more natural features, both in terms of body and background appearance, should enhance the N190 amplitude.

## Methods

**Participants.** The same participants who were recruited for Experiment 1 also took part in Experiment 2. As such, the demographic information of this sample is identical to that reported previously. Participants completed Experiments 1 and 2 in a counterbalanced order, with half of the participants completing Experiment 1 first and the remainder completing Experiment 2 first.

**Stimuli.** We used additional images from the Penn-Fudan database for this second experiment, with a few key differences. First, we limited ourselves to images containing a single pedestrian rather than allowing body numerosity to vary systematically as in Experiment 1. Second, we used the segmentation masks described above in Experiment 1 to create two alternate versions of each original image: (1) As in Experiment 1, the segmentation mask was applied to render each pedestrian as a silhouette, and (2) The same mask was applied to remove the background of the image while preserving the natural appearance of the pedestrian in the scene (Fig 7). We therefore varied natural appearance of the body and the background across three conditions, a "silhouette" condition depicting no background and only a silhouette of a pedestrian's body, a "natural body" condition depicting no background but photographic detail of a pedestrian's body, and finally a "natural context" condition including both a

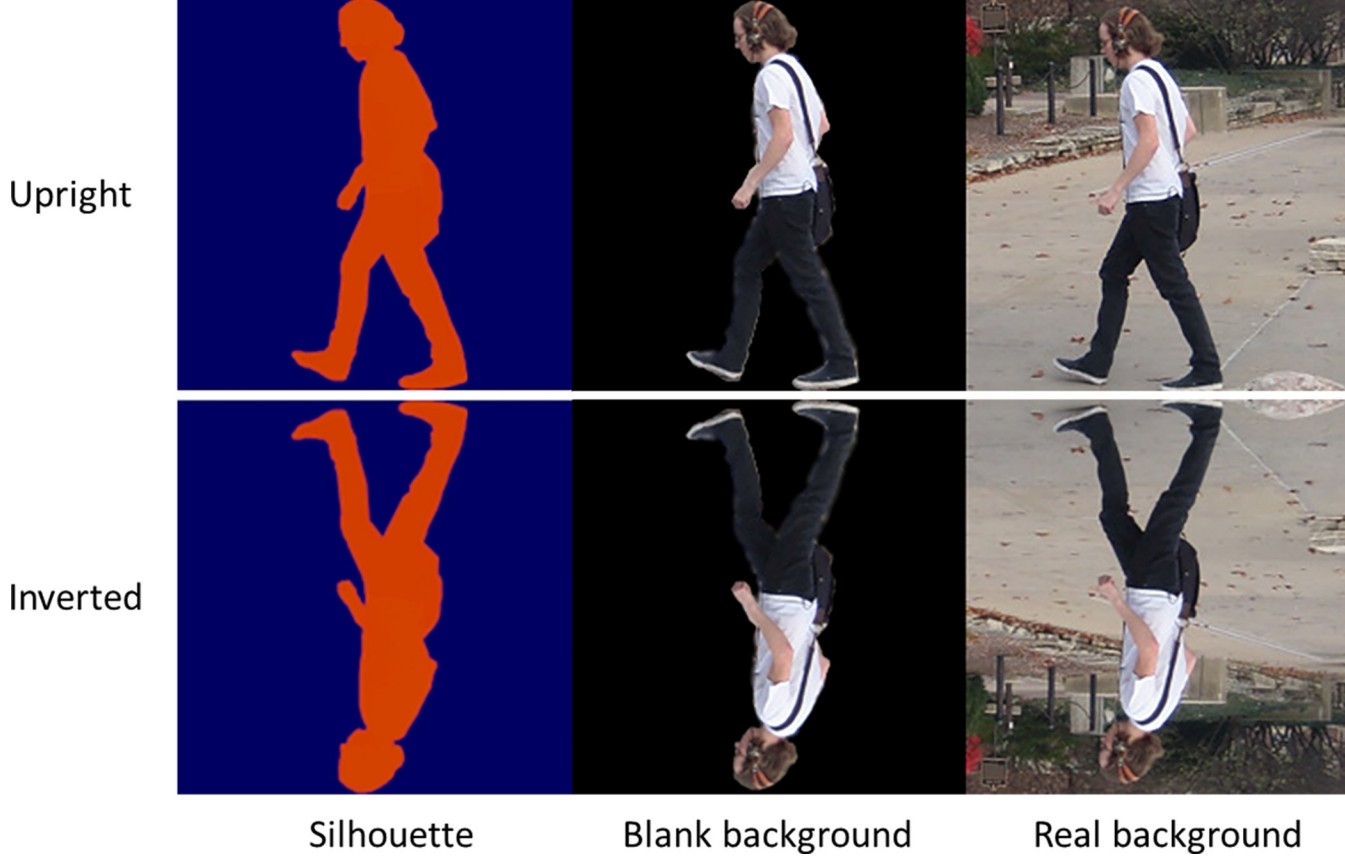

**Fig 7. Example images similar to stimuli from Experiment 2.** We used metadata from the Penn-Fudan database to selectively remove the presence of natural body and/or scene texture, yielding a silhouette-only condition, a black background condition where only the individual in the scene has natural appearance, and a real background condition that includes foreground and background texture. As in Experiment 1, all images were presented in an upright and inverted (vertically flipped) orientation in separate conditions. Due to the variability in body position across images, position of the body in the upper or lower half of the visual field was not systematically confounded with orientation.

full photographic background and photographic detail of a pedestrian's body. Additionally, all images were further presented in both an upright and an inverted orientation. We selected 64 images per condition for this experiment, with the same images presented in the upright and inverted conditions. We did not control for the visibility of individual faces in images.

**Procedure.** All testing procedures and EEG protocols were identical to those described in Experiment 1. Participants carried out an orientation categorization task during EEG recording, signalling their response via button box. Each stimulus image was presented once during the recording session for a grand total of 384 trials in the entire experiment. Each recording session lasted approximately 25 minutes.

**EEG preprocessing.** We applied the same preprocessing steps to the continuous EEG data collected in Experiment 2 as described previously in Experiment 1. The same sensors of interest were determined to be appropriate in this analysis based on visual inspection of the grand average ERP scalp distribution collapsed across participants and conditions. To analyze the N190 amplitude and latency, we selected a time window of 144ms - 216ms based on inspection of the grand average ERP collapsed across participants at the sensors of interest. This interval was sufficient to capture the N190 component across participants and conditions (Fig 8).

## Results

**Behavioral performance.** As in Experiment 1, across all conditions participants' accuracy at reporting the orientation of the stimuli (upright or inverted) exceeded 94% with response latencies to correct responses under 650ms for all conditions. Again, because performance was clearly at ceiling for this task we do not include further discussion of this data.

**N190 amplitude.** Within the time interval described above we measured the mean amplitude per participant for each condition (Fig 9) and analyzed these values with a 3x2x2 repeated-measures ANOVA with appearance (body silhouette, body photograph, and photographic background), orientation (upright vs. inverted) and hemisphere (left vs. right) as within-subjects factors. This analysis revealed a main effect of appearance ($F_{(2,30)} = 41.57$, $p<0.001$, $\eta^2_p = 0.74$) which was the result of significant differences between all three levels of appearance (Cohen's $d>1.5$ for all post-hoc tests, $p_{bonf}<0.001$). We also observed significant interactions between orientation and hemisphere ($F_{(1,15)} = 4.98$, $p = 0.041$, $\eta^2_p = 0.25$) and orientation and appearance ($F_{(2,30)} = 4.78$, $p = 0.016$, $\eta^2_p = 0.24$). With regard to the interaction between orientation and hemisphere, post-hoc tests (corrected for a family of 6 comparisons) revealed only a marginal difference between upright and inverted responses observed in the right hemisphere in the same direction as the inversion effect in Experiment 1. ($t = 2.58$, Cohen's $d = 0.18$, $p = 0.09$). Regarding the interaction between orientation and appearance, post-hoc tests (corrected for a family of 15 comparisons) revealed a complicated pattern of results. We have included the full table of post-hoc pairwise comparisons below (Table 1), but a key result is that we only observe a significant inversion effect for the Regular Background appearance condition ($t = 2.85$, Cohen's $d = 0.26$, $p = 0.026$), while neither of the other appearance conditions exhibit a significant inversion effect.

## N190 peak latency

Within the critical time interval we also measured the peak latency by determining the time point associated with the most negative amplitude for each participant across all conditions. We analyzed these results using an ANOVA with the same structure described above for our analysis of the mean amplitude. This analysis revealed a marginal main effect of orientation ($F_{(1,15)} = 23.8$, $p<0.069$, $\eta^2_p = 0.20$) and a significant interaction between orientation and appearance ($F_{(2,30)} = 3.66$, $p<0.038$, $\eta^2_p = 0.20$). Post-hoc tests (corrected for a family of 15 comparisons revealed that in fact no pairwise comparisons reached significance.

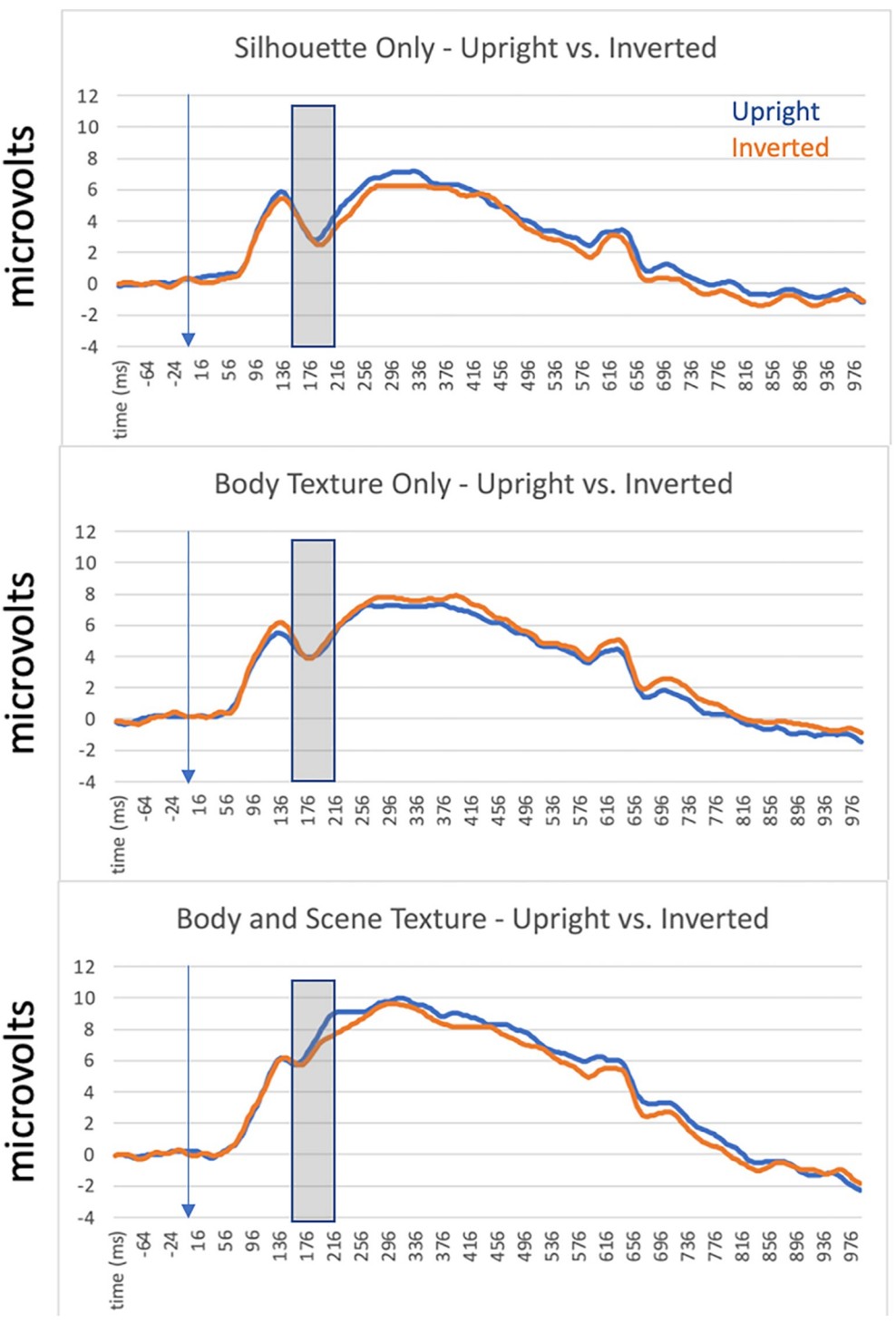

**Fig 8. Grand average ERPs calculated across participants in all conditions in Experiment 2.** These data arecollapsed across left and right hemisphere sensors to highlight the effects of image orientation and natural foreground and background texture. The transparent grey window indicates the approximate time window used to analyze the N190 component.

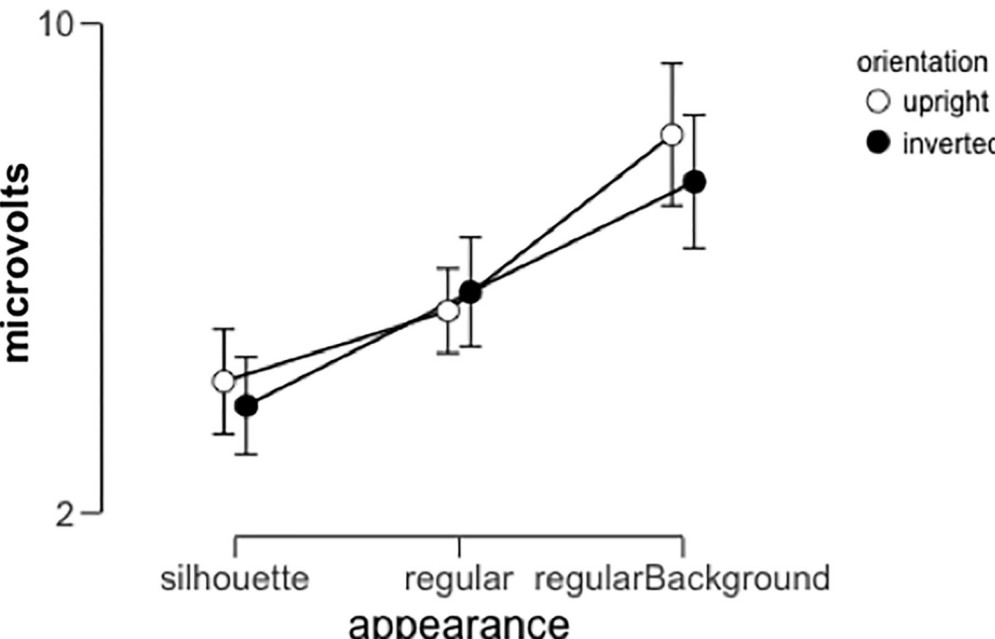

**Fig 9. Average N190 amplitude across participants as a function of image orientation (upright or inverted) and appearance conditions (silhouette only, regular body appearance, regular background included).** Error bars represent 95% confidence intervals.

## Decoding analysis

Decoding accuracy of all pairwise condition comparisons was above chance level (Fig 10). The overall decoding accuracy was also higher as compared to Exp. 1 (average accuracy of 63% vs

**Table 1. Post-hoc pairwise tests to examine the interaction between orientation and body/scene appearance.**

Post Hoc Comparisons—orientation ✳ appearance

| | | Mean Difference | SE | t | Cohen's d | $p_{holm}$ |
|---|---|---|---|---|---|---|
| upright, silhouette | inverted, silhouette | 0.396 | 0.269 | 1.470 | 0.133 | 0.298 |
| | upright, regular | -1.155 | 0.461 | -2.509 | -0.388 | 0.049 |
| | inverted, regular | -1.464 | 0.471 | -3.105 | -0.492 | 0.017 |
| | upright, regularBackground | -4.035 | 0.461 | -8.761 | -1.356 | < .001 |
| | inverted, regularBackground | -3.267 | 0.471 | -6.930 | -1.098 | < .001 |
| inverted, silhouette | upright, regular | -1.551 | 0.471 | -3.290 | -0.521 | 0.012 |
| | inverted, regular | -1.859 | 0.461 | -4.037 | -0.625 | 0.002 |
| | upright, regularBackground | -4.430 | 0.471 | -9.398 | -1.489 | < .001 |
| | inverted, regularBackground | -3.662 | 0.461 | -7.952 | -1.231 | < .001 |
| upright, regular | inverted, regular | -0.308 | 0.269 | -1.146 | -0.104 | 0.298 |
| | upright, regularBackground | -2.879 | 0.461 | -6.252 | -0.968 | < .001 |
| | inverted, regularBackground | -2.111 | 0.471 | -4.479 | -0.710 | < .001 |
| inverted, regular | upright, regularBackground | -2.571 | 0.471 | -5.454 | -0.864 | < .001 |
| | inverted, regularBackground | -1.803 | 0.461 | -3.915 | -0.606 | 0.002 |
| upright, regularBackground | inverted, regularBackground | 0.768 | 0.269 | 2.854 | 0.258 | 0.026 |

*Note.* P-value adjusted for comparing a family of 15.

*Note.* Results are averaged over the levels of: Hemisphere.

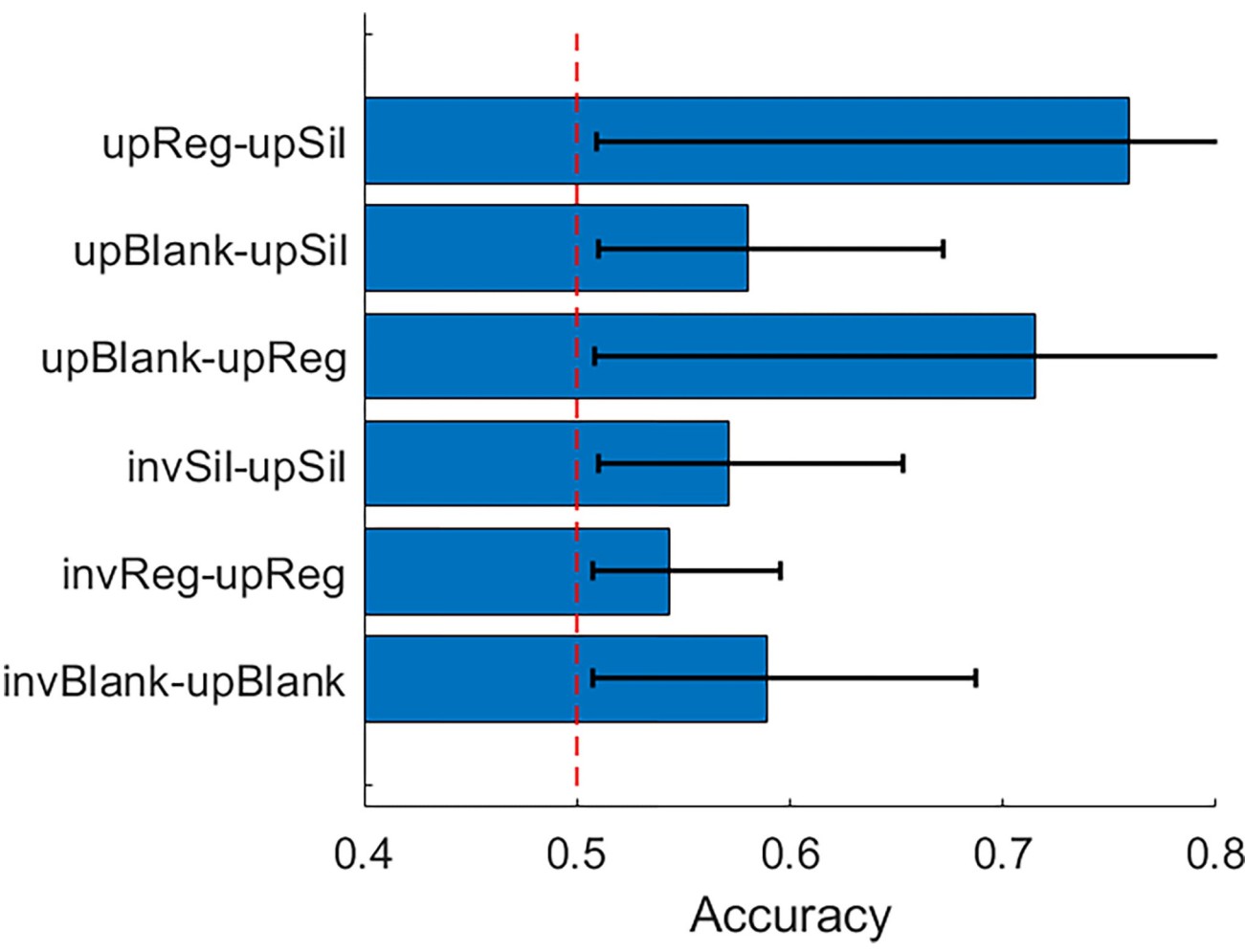

**Fig 10. Pairwise decoding accuracy for all comparisons in Experiment 1.** Up: Upright; Inv: Inverted; Blank: Blank body and background appearance; Sil: Blank body appearance, regular background appearance; Reg: Regular body and background appearance. Error bars represent 95% CIs.

57%), suggesting that body/background appearance modifications modify ERP responses to a greater extent than the body numerosity. We observed A similar pattern of spatiotemporal feature contributions as in Exp. 1 for the inversion comparisons,where features other than those around N190 were assigned higher ranking when comparing conditions (Fig 11). However, for the appearance changes, we were also able to detect a spread of increased feature diagnosticity over the posterior electrodes at around N190 and up to 200ms after, as opposed to the more diffuse increase of feature diagnosticity in Exp. 1 (Fig 12). This may mean that manipulating the body and scene appearance variables in this task had a larger impact on early visual responses than the manipulation of numerosity did, which is consistent with the more dramatic changes in low-level image properties like orientation energy and contrast that accompany the removal of object texture and background objects and surfaces.

## General discussion

Our results from Experiment 2 complement and extend several of the outcomes we observed in Experiment 1. First, in this second task we were able to obtain more typical-looking N190 components in both conditions that lacked a complex background. Given that the same bodies

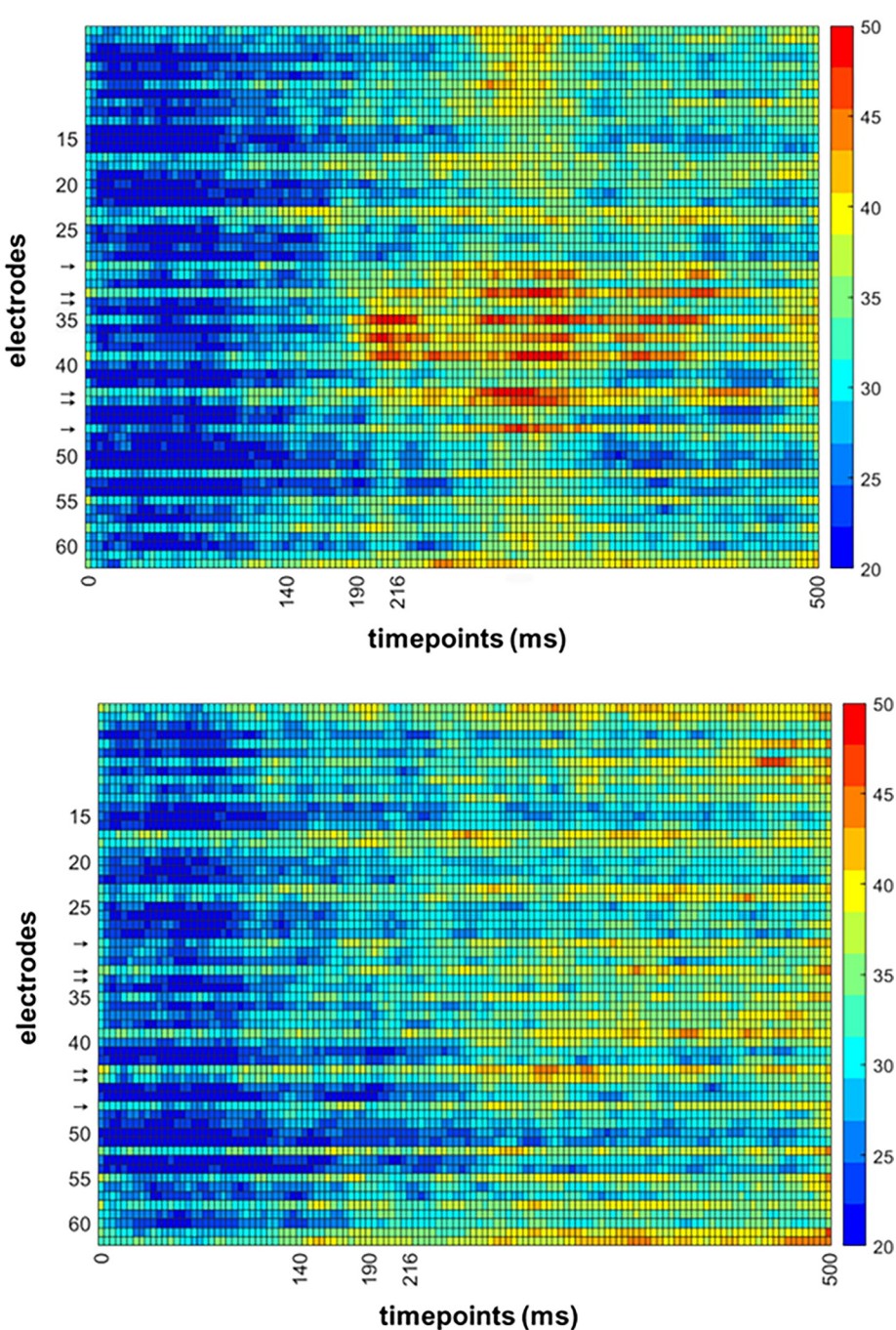

**Fig 11. a.** Average ranking of spatiotemporal features (electrode x timepoint) for decoding body and scene appearance across all pairwise comparisons and collapsed across upright conditions. Inverted conditions were not included in this analysis. Warmer colors represent spatiotemporal bins that were ranked higher on average across all iterations of our classification analysis. We find many highly ranked features in the same spatial and temporal neighborhood as the N190, but again find that multiple other bins contain diagnostic information about the appearance of bodies and scene background. → = Electrodes used in the ERP analysis. **b.** Average ranking of spatiotemporal features (electrode x timepoint) for decoding orientation, collapsed across all matched appearance conditions (Upright Silhouette vs. Inverted Silhouette, e.g.). Warmer colors represent spatiotemporal bins that were ranked higher on average across all iterations of our classification analysis. In this case, highly ranked features for orientation are less evident in the N190 spatiotemporal neighborhood, but once again we observe other bins that carry diagnostic information for decoding body inversion. → = Electrodes used in the ERP analysis.

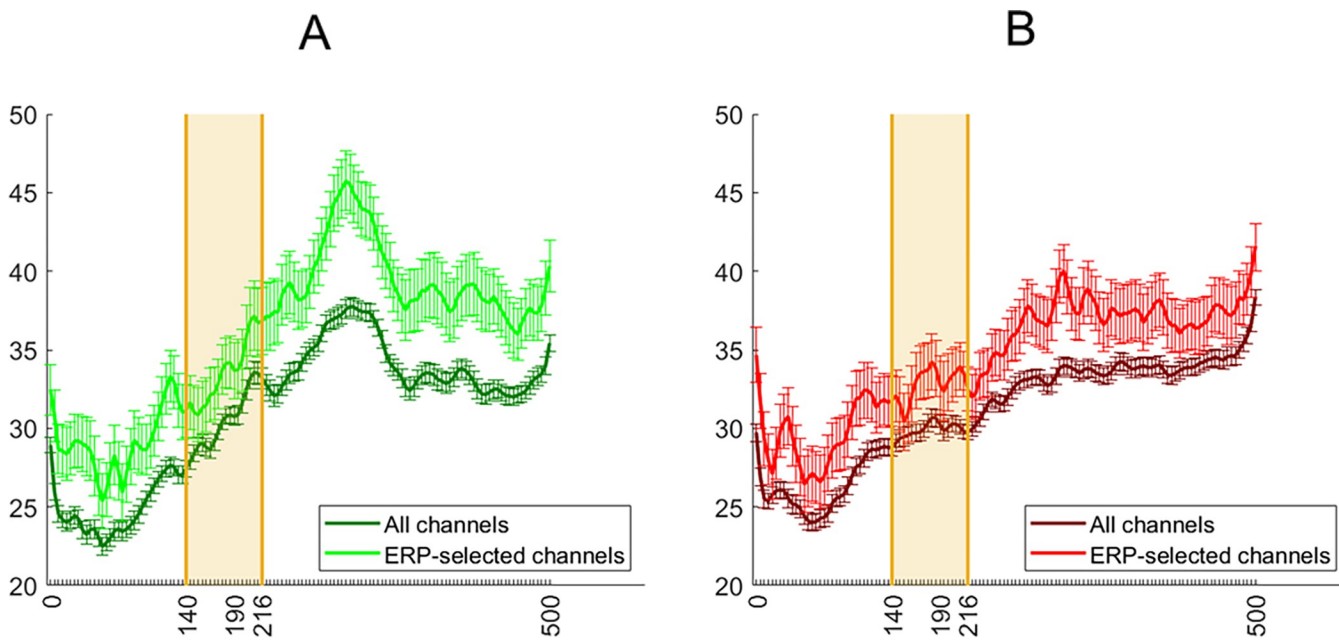

**Fig 12.** Ranking of temporal features, averaged across electrodes for decoding appearance between upright conditions (A), and decoding orientation in matching appearance conditions (B). Orange shading represents the chosen interval around N190. Again, while there are moderately highly-ranked features in the same spatial and temporal neighborhood as the N190, there are also multiple additional features that were also ranked highly.

were depicted in all of our stimulus conditions in this task, we interpret this as evidence that the presence of the natural context in Experiment 1 was the driving factor in reducing the overall amplitude of the N190. While natural backgrounds could provide useful context to bolster the response to images of bodies, in the stimuli used in this task it seems that rich backgrounds served more as clutter than as support for body processing. A key limitation of our study, however, is whether or not it is the presence of multiple recognizable objects, segmentable surfaces, and potentially a nameable scene that affects the N190 component, or the presence of low-level clutter. Both conditions in Experiment 2 that only included the body in each image had completely blank backgrounds, obviating the need to segment the object from the background. Replacing the natural background with either a synthetic texture matched to the statistics of the original scene [53] or a warped background that similarly preserves low-level image structure [54] would be an important next step to determine what aspects of natural scene structure contribute to the reduction of the N190 that we have observed across our two tasks.

The body inversion effects we observed in Experiment 2 complicate our interpretation, however. While we observed consistent "inverted" inversion effects in Experiment 1, we observed relatively small effects of inversion in Experiment 2 that are more consistent with prior reports describing a more negative N190 amplitude in response to inverted bodies. It is obviously challenging to account for this discrepancy between Experiments 1 and 2, especially given the similarity in the images used across the two tasks. The absence of a measurable inversion effect in our "Regular Body Appearance" condition is especially puzzling because these images depicting a detailed body on a blank background most closely resemble the results of many prior studies using constrained stimuli. While it may be tempting to speculate on potential reasons why we may have observed this pattern of results, we think it more prudent to emphasize two points, one related to the larger literature describing the body inversion effect

and another related to our characterization of the N190. First, regarding prior reports describing inversion effects at the N190, a recent meta-analysis [55] highlighted the variability in methods and analyses in the ERP literature describing this effect. While they conclude from their meta-analysis that the BIE is a moderately-sized effect, they also determined that it is smaller than the typical effect size for the face inversion effect. This latter observation is potentially important to consider alongside results suggesting that the behavioral body inversion effect may largely be mediated by face-sensitive cortical areas rather than body-sensitive loci [56]. Thus, the nature of the body inversion effect may be multiply determined by aspects of face visibility and appearance, body appearance and posture, and the context in which the body is depicted. The second point we think it is worthwhile to emphasize is that while the N190 amplitude results across our two experiments were variable, we did observe more consistent effects of body inversion on N190 latency, consistent with prior reports (longer latency-to-peak for inverted bodies). The diffuse nature of the N190 peaks that we observed in Experiment 1 complicate this observation somewhat, but still support consistent effects of orientation on the N190 component that do not interact with numerosity (Experiment 1) but do interact with natural body and scene appearance (Experiment 2).

The low-amplitude and temporally diffuse N190 components that we observed in Experiment 1 are challenging to characterize using window-based methods for isolating the spatial and temporal extent of ERP components. In both experiments, our decoding analyses offer some intriguing additional insights into the nature of the stimulus effects that we observed in both of our tasks. Overall, the most important conclusion from this analysis is that neural sensitivity to body orientation, body and scene appearance, and body numerosity extend beyond the N190 component both spatially and temporally. Some of the highly-weighted features revealed by our decoding analysis are difficult to interpret in terms of recognition mechanisms: The high weights assigned to our eye electrodes, for example, are unlikely to reflect the contribution of visual areas. Though we ultimately excluded these features from our decoding analyses, the data is potentially intriguing in that it may indicate that there are differences in eye movement planning or execution as a function of our stimulus parameters, which in itself is an interesting topic for future study. These features of the data aside, our visualizations of the highest-weighted spatiotemporal features for decoding orientation and appearance variables also indicate that there is information available for reading out these aspects of our stimuli outside the temporal window for the N190. To take a broad view of these results, we conclude that especially with regard to bodies appearing in complex scenes with other people, other objects and rich backgrounds, visual processing of the body may be best characterized in terms of a distributed response. Continued use of tools for mass univariate analyses [57,58] and neural decoding is likely to reveal that sensitivity to body and scene appearance is manifest at multiple stages of processing.

Overall, our results indicate that the neural response to bodies is modulated by multiple aspects of body appearance that are directly related to how bodies typically appear in natural settings. That is, when we use images that are more like observers' experience of seeing bodies "in the wild" the characteristics of the N190 are different than when we use more limited stimuli. In our two experiments, we make the case for these effects using a set of largely unconstrained images with what could be considered ambient variability in body appearance. The study of ambient variability in facial appearance has emerged as a key topic in face recognition research [59] and our results have a kinship with those insofar as in both cases the key initial observation is recognizing the impact of natural variability. That is, we have made no attempt to systematically vary or quantify the position, size, or posture of the bodies in our study, but instead treat this unconstrained variability as a useful feature of our task rather than as a nuisance. While systematic study of these factors will be a useful direction for ongoing work, our

results provide a useful demonstration that these stimulus factors do matter. To push a little further along these lines, analyzing ERP data at the level of single trials complemented by meta-data from labeled complex images [60] could be a useful way to leverage natural variability across images to identify the dimensions of body sensitivity in different neural responses. This approach would likely require large amounts of data from individual participants, but would also offer important insights into how body processing works in real-world settings. At present, the current study offers important indications that variation in the appearance of bodies in natural scenes can have large consequences for one of the key neural indices of body-specific processing.

## Acknowledgments

Our thanks to Dr. Jeff Boyer for facilitating recruitment and testing of North Dakota State Governor's School participants in the experiments described here. We would also like to thank Dan Gu and Ganesh Padmanabhan for technical assistance that supported this work.

## Author Contributions

**Conceptualization:** Abigail Post, Benjamin Balas.

**Data curation:** Benjamin Balas.

**Investigation:** Abigail Post, Alyson Saville.

**Methodology:** Ilya Nudnou, Abigail Post, Alyson Saville, Benjamin Balas.

**Project administration:** Alyson Saville.

**Software:** Ilya Nudnou, Benjamin Balas.

**Supervision:** Alyson Saville.

**Visualization:** Ilya Nudnou.

**Writing – original draft:** Ilya Nudnou, Benjamin Balas.

**Writing – review & editing:** Ilya Nudnou, Abigail Post, Benjamin Balas.

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
