## [Decision Letter · Decision Letter 0]

4 Jan 2023

PONE-D-22-31727Putting People in Context: ERP Responses to Bodies in Natural ScenesPLOS ONE

Dear Dr. Balas,

Thank you for submitting your manuscript to PLOS ONE. After careful consideration, we feel that it has merit but does not fully meet PLOS ONE’s publication criteria as it currently stands. Therefore, we invite you to submit a revised version of the manuscript that addresses the points raised during the review process.

Two expert reviewers have evaluated your study. Both reviewers found your work interesting and well executed, but both also raise important points that need to be addressed. Reviewer 1 in particular notes the lack of a control category, and both reviewers ask for important clarifications regarding the methods and the interpretation of your results. The reviewers also provide helpful and constructive suggestions to address these points, so I believe it should be possible for you to successfully address all the reviewer comments, and I look forward to receiving your revisions. 

We look forward to receiving your revised manuscript.

Kind regards,

Guido Maiello

Academic Editor

PLOS ONE

Journal Requirements:

4. Please ensure that you refer to Figures 6 and 7 in your text as, if accepted, production will need this reference to link the reader to the figure.

5. We note that Figures 1 and 6 include images of participants in the study. 

Reviewers' comments:

Reviewer's Responses to Questions

**Comments to the Author**

1. Is the manuscript technically sound, and do the data support the conclusions?

Reviewer #1: Yes

Reviewer #2: Yes

2. Has the statistical analysis been performed appropriately and rigorously? 

Reviewer #1: Yes

Reviewer #2: Yes

3. Have the authors made all data underlying the findings in their manuscript fully available?

Reviewer #1: Yes

Reviewer #2: Yes

4. Is the manuscript presented in an intelligible fashion and written in standard English?

Reviewer #1: Yes

Reviewer #2: Yes

5. Review Comments to the Author

Reviewer #1: Review of PLOS ONE 22-31727

This paper presents two EEG studies investigating the effect of body inversion on the previously reported body-selective EEG component (N190). The focus is on how the inversion effect is influenced by the number of bodies (Exp 1) and the presence of scene background (Exp 2). The studies are well-conducted and address novel questions. The paper is also easy to follow. However, I have several major concerns:

-The study did not include a control category. This makes it difficult (if not impossible) to directly compare the N190 amplitudes across stimulus conditions because the evoked response between 140-216 ms (taken as the N190 here) will be affected by any difference in visual input. For example, the difference between bodies with vs without background could be driven by the background itself. Similarly, the difference between one vs multiple bodies could be driven more generally by increased clutter. For this reason, the main comparisons in the manuscript could focus on the inversion effect, and how this inversion effect differs across conditions. I would encourage the authors to make this focus clear from the beginning (starting with the hypotheses) and avoid comparing responses across categories as these may not be related to body processing as such.

-The definition of the N190 (e.g., 140-216 ms) was not sufficiently motivated. The text states that this was based on inspection, but it is not clear what exactly determined this precise time window.

-Because there was no control category, it is unclear whether the N190 (as defined here) was body selective and corresponded to the previously reported body-selective N190 (e.g., Thierry et al., 2006; Taylor et al., 2010). Please comment on the correspondence of the N190 here and in previous reports, both based on selected electrodes and definition of time window. Ideally, additional analyses would be included that more closely follow the N190 as defined previously, particularly because results are not always in line with previous reports.

-The sample size (N=16) is on the low end for today’s standards. Please provide information about how the sample size was determined (e.g., was a power analysis conducted?).

-How was inversion achieved? Were the images rotated by 180 degrees? Please include the inverted versions in the stimulus example figures (Fig 2 and Fig 6) so that it is immediately clear what the main manipulation was. Did the conditions differ in the visual field that was stimulated (e.g., Exp 2: upright: most of the body in lower visual field, inverted: most of the body in upper visual field). If so, could that explain the differences between the effects observed here and in previous literature?

-Some papers that may be relevant to cite or discuss:

Related to the neural basis of body and face perception:

FFA: Kanwisher et al., J Neurosci 1997; EBA: Downing et al., Science 2001; FBA: Peelen & Downing, J Neurophys 2005

Related to the body-selective N1:

Gliga & Dehaene-Lamertz, J Cogn Neurosci 2005; Pourtois et al., Neuropsychologia 2007; Taylor et al., Brain & Cogn 2010

Related to body processing in natural scenes:

Bindemann et al. JEP:HPP 2010; Kaiser et al., J Neurosci 2016

-Please provide the behavioral results (accuracy and RT) for both experiments; did these differ across conditions?

-When reporting the statistics, please also report the non-significant interactions that are relevant to the research question (e.g., on p.16, the interaction between orientation and numerosity).

-For the Latency analysis, please provide the means, either in the text or in a figure (e.g., in the format of Figure 3, possibly as panel B in that figure; same for Experiment 2).

-For the decoding analysis, it was unclear why such a long time-window was included (up until 1s after stimulus onset). This time window presumably includes the response of the participant, which was directly reflecting stimulus orientation. Considering that the focus of the study is on the visual processing of bodies, I would limit the time window to the visually evoked response (e.g., 0-400 ms).

-The significant interactions (e.g., on p.29) should be followed up by additional tests (e.g., separate 2x2 ANOVAs or pairwise t-tests) to interpret these interactions. For example, for the data shown in Fig 8 it is not clear which differences drove the 3x2 interaction.

Minor:

-p.7: “affects of wider-ranging”  “effects of wider-ranging”

-Fig 1: The examples stand out in not showing the inner facial features of the people. Was this on purpose, and true for most of the images in the database, or was this specific to these examples?

-The beginning of the Results section (about preprocessing etc.) would fit better in the Methods section.

-p.14: Please provide the correspondence between the electrode numbers and standard electrode labels; e.g., did these include P7/P8?

-Fig 2 and Fig 7 miss y-axis legends and legends for the conditions (colored lines). The x-axis is hard to read, also because there is no clear 0 point. It would be informative to indicate the N190 window using shading or similar.

-Fig 2 legend “This data is”  “These data are”

Reviewer #2: In this paper, the Authors explore how the N190 (ERP component related to body processing) is affected by several, previously unexplored, aspects related to person perception, generally linked to the stimuli’s ecological validity (few/multiple bodies – presence of a real world scene in the background). While in terms of results there are some limitations (due for example to the inconsistency of results related to inversion effects between experiment 1 and 2, and previous literature), the Authors take careful/appropriate conclusions based on their findings. Indeed, natural appearance of the body “in the wild” seem to matter for the N190 amplitude, and for decoding accuracy in general. A side conclusion is also that the N190 cannot be fully accounted for by low-variability within the stimuli set as argued in the past (Thierry et al., 2007).

I think that the manuscript is overall well written, particularly in the intro and discussions, and I enjoyed reading it.

However, I think the paper would benefit with some revisions:

Could several of the effects observed on the N190 be accounted by attentional deployment to body stimuli, rather than some genuine differences due to configural processing? – For example, N190 was smaller (more negative) with multiple people, and more positive with upright stimuli (as opposed to inverted). Both conditions more than one person, and the “weirdness” of an inverted body, supposedly increase the likelihood that someone would gaze to a body/person within a scene, therefore influencing the N190. Generally, it would be useful if in the results the Authors were also consistent using “more negative” or “more positive” – see also points below.

Is that the case that the Authors expected an interaction with the effect of one – to – many people modulating the N190 as a function of inversion? Seems so from what stated in the abstract. If that is the case, could the Author explain why such interaction might be expected?

Hemisphere by orientation interactions (both in experiment 1 and 2) – I found these interactions quite intriguing because if inversion affects configural processing, then inverted stimuli body part-based processing should be increasing. Single body part processing seems to rather involve more the left hemisphere (Urgesi et al., 2007; Bracci et al., 2012). Here you instead stronger inversion effects on the right, perhaps reinforcing the interpretation that in the case of the N190 the inversion effect is rather explained by domain general attentional effects rather than configural processing as such.

I found the results of the classification analyses a bit hard to read. Is quite clear what you did – but less clearly written the conclusions you take from them. For example, in the abstract you mention that orientation decoding is also contributed later (after 300ms). However, this is not clearly shown in the result sections. I appreciate the figure 4a and 4b such that all the electrodes can be seen, however, it would be important perhaps also to show a line graph with decoding accuracy on the y axis and time in the x axis across all the electrodes (and/or only taking the electrodes of interest, e.g. those of the N190 or all the posterior electrodes?). In this way would be clearer the “when” of orientation/numerosity/appearance classification reaches significance against chance. Also – do you think would be useful to look at the decoding accuracy of numerosity (exp 1) and appearance (experiment 2), as presented in the figure 4a and 10a without averaging across upright and inverted, and only looking at the upright condition? Of course orientation was task relevant and did not show interactions, but this may be still interesting to look at?

The paper would really benefit with some more detailed data visualization, for example it can be quite confusing, particularly for a non-eeg expert reader, to deal with expressions like “more positive”/”less negative” amplitude in one or the other condition. – see minor points specific on the figures.

Minor:

E1 Stimuli:

Nice database, what is this database usually used for? Computer vision?

In the stimuli example in Figure 1 the face was always non- or barely- visible – is that something that you controlled for, or in some cases the face was fully visible? Would be useful to specify this since, as the Authors acknowledge, that might matter for the inversion effects.

Figure 2 and Figure 7 do not contain any legend for upright and inverted, also is not entirely clear what the y axis represent, because the labels are absent. I am not entirely sure also what was the purpose of showing these figures?

Figure 3 – Would be nice that the figure would be a line plot showing the actual component also, would make the results sections clearer – instead of trying to read through “more negative, more positive amplitude”. Also in the captions it says “across conditions” but it should rather be “in each condition”.

Page 16 – results: Indicate also the contrast between few and many even if it didn’t reach significance; In this section would be also useful to have descriptives related to the differences (Mean values and 95% CIs so that we could make a sense of the extent of the effects.

Figure 4a. is this collapsed across upright and inverted conditions or represent only the upright condition? Seems that there is a typo in the caption.

Discussion Exp 1 and 2: Would be interesting to know whether the inconsistencies between studies on inversion effects affecting the N190 relate to the task-relevance this factor had in the experiments?

Results Experiment 2: N190 amplitude: Please include the relevant contrasts at the end when you break down the two significant interactions.

6. PLOS authors have the option to publish the peer review history of their article (what does this mean?). If published, this will include your full peer review and any attached files.

Reviewer #1: No

Reviewer #2: No

---

## [Author Response · Author response to Decision Letter 0]

15 Feb 2023

Response to reviews:

We would like to thank our reviewers for their comments on our initial submission. We found their suggestions to be very helpful and we have edited the text to address the points that they raised. Below, we reproduce their reviews in their entirety and indicate how we have responded to their comments point-by-point. The original reviews are in plain text and our responses are in bold.

First, regarding Figures 1 and 6, we wish to clarify for the Journal Requirements that these are not images of participants, but images from the Fudan Database, which was the stimulus set used in our tasks. Our understanding based on the terms of use for that database is that these images are available for use in published work with the database. 

Reviewer #1: Review of PLOS ONE 22-31727

This paper presents two EEG studies investigating the effect of body inversion on the previously reported body-selective EEG component (N190). The focus is on how the inversion effect is influenced by the number of bodies (Exp 1) and the presence of scene background (Exp 2). The studies are well-conducted and address novel questions. The paper is also easy to follow. However, I have several major concerns:

-The study did not include a control category. This makes it difficult (if not impossible) to directly compare the N190 amplitudes across stimulus conditions because the evoked response between 140-216 ms (taken as the N190 here) will be affected by any difference in visual input. For example, the difference between bodies with vs without background could be driven by the background itself. Similarly, the difference between one vs multiple bodies could be driven more generally by increased clutter. For this reason, the main comparisons in the manuscript could focus on the inversion effect, and how this inversion effect differs across conditions. I would encourage the authors to make this focus clear from the beginning (starting with the hypotheses) and avoid comparing responses across categories as these may not be related to body processing as such.

We agree with the reviewer that our inversion manipulation is an important control for low-level stimulus properties. That said, we also think that it is meaningful to examine how variation in such properties affects the response of putative body-sensitive ERP components. This latter goal does not lend itself to conclusions that rely on isolating body processing from other aspects of visual recognition, but nonetheless contributes to our understanding of how neural responses may play out in natural settings where there is not precise control over stimulus appearance. We have edited the text to emphasize the analysis of interactions with inversion in our data and also highlight that one of our goals is to see how natural scene context may affect the responses we measure to images with bodies in them. With regard to the latter point, we emphasize that such effects cannot be linked directly to body processing in the specific, but rather reflect the impact of multiple stages of processing on the responses that are measured to body images.

-The definition of the N190 (e.g., 140-216 ms) was not sufficiently motivated. The text states that this was based on inspection, but it is not clear what exactly determined this precise time window.

We apologize for the lack of clarity - in the revised text we outline our procedures for identifying the time window of interest.

-Because there was no control category, it is unclear whether the N190 (as defined here) was body selective and corresponded to the previously reported body-selective N190 (e.g., Thierry et al., 2006; Taylor et al., 2010). Please comment on the correspondence of the N190 here and in previous reports, both based on selected electrodes and definition of time window. Ideally, additional analyses would be included that more closely follow the N190 as defined previously, particularly because results are not always in line with previous reports.

Similar to our previous comment, body inversion serves as a control category in this study and we have revised the text to emphasize that the body inversion effect serves as a means of confirming that we are measuring a component that is to some extent specific to body processing. With regard to the sensors and time windows of interest, we have included more information about the correspondence between our parameters and previous reports. It was not clear to us which additional analyses the reviewer thought would be more informative, so the present draft does not include further analysis of this component. 

-The sample size (N=16) is on the low end for today’s standards. Please provide information about how the sample size was determined (e.g., was a power analysis conducted?).

We determined the sample size based on a power analysis that relied on previous effect sizes related to the body inversion effect. In particular, we note that this sample size is larger than the N of 12 used in Downing et al’s original report of the N190, which included an analysis of the inversion effect. 

-How was inversion achieved? Were the images rotated by 180 degrees? Please include the inverted versions in the stimulus example figures (Fig 2 and Fig 6) so that it is immediately clear what the main manipulation was. Did the conditions differ in the visual field that was stimulated (e.g., Exp 2: upright: most of the body in lower visual field, inverted: most of the body in upper visual field). If so, could that explain the differences between the effects observed here and in previous literature?

Inversion was achieved by flipping images vertically. This did not reliably place the body in the upper or lower visual field as there was substantial variability across images in the position of the body in each scene. The idea that this variability (rather than a systematic bias in our images) could help account for the differences we observe is a good point and one we think our data is consistent with. We have revised the text to highlight this possibility. 

-Some papers that may be relevant to cite or discuss:

Related to the neural basis of body and face perception:

FFA: Kanwisher et al., J Neurosci 1997; EBA: Downing et al., Science 2001; FBA: Peelen & Downing, J Neurophys 2005

Related to the body-selective N1:

Gliga & Dehaene-Lamertz, J Cogn Neurosci 2005; Pourtois et al., Neuropsychologia 2007; Taylor et al., Brain & Cogn 2010

Related to body processing in natural scenes:

Bindemann et al. JEP:HPP 2010; Kaiser et al., J Neurosci 2016

Thank you for these - we have incorporated these into the revised text.

-Please provide the behavioral results (accuracy and RT) for both experiments; did these differ across conditions?

We have done so, but we also note that the behavioral task (signal upright/inverted orientation) is very easy and served primarily as a simple attention check to ensure that participants were attending to our stimuli throughout the task. As such, performance was uniformly high across condition (mean accuracy over 96% across all conditions) and RTs were similarly fast. 

-When reporting the statistics, please also report the non-significant interactions that are relevant to the research question (e.g., on p.16, the interaction between orientation and numerosity).

We have done so in the revised text. 

-For the Latency analysis, please provide the means, either in the text or in a figure (e.g., in the format of Figure 3, possibly as panel B in that figure; same for Experiment 2).

We have done so in the revised text. 

-For the decoding analysis, it was unclear why such a long time-window was included (up until 1s after stimulus onset). This time window presumably includes the response of the participant, which was directly reflecting stimulus orientation. Considering that the focus of the study is on the visual processing of bodies, I would limit the time window to the visually evoked response (e.g., 0-400 ms).

Thank you for pointing this out. Indeed, stimulus offset is at 500 ms, which means any further features may include non-perceptual components such as the motor response. We re-ran the decoding analyses to include only timepoints from 0-500 ms, and updated all existing figures with the new results. We also added additional interpretation where necessary.

-The significant interactions (e.g., on p.29) should be followed up by additional tests (e.g., separate 2x2 ANOVAs or pairwise t-tests) to interpret these interactions. For example, for the data shown in Fig 8 it is not clear which differences drove the 3x2 interaction.

We have included this in the revised text.

Minor:

-p.7: “affects of wider-ranging”  “effects of wider-ranging”

Modification made, thank you.

-Fig 1: The examples stand out in not showing the inner facial features of the people. Was this on purpose, and true for most of the images in the database, or was this specific to these examples?

We did not control the visibility of the face and acknowledge this in the revised text as suggested here. The examples were chosen primarily to be a good representation of the experimental conditions.

-The beginning of the Results section (about preprocessing etc.) would fit better in the Methods section.

We have made this adjustment, thank you. 

-p.14: Please provide the correspondence between the electrode numbers and standard electrode labels; e.g., did these include P7/P8?

Thank you for the suggestion, we have included the approximate 10-20 coordinates in parentheses. 

-Fig 2 and Fig 7 miss y-axis legends and legends for the conditions (colored lines). The x-axis is hard to read, also because there is no clear 0 point. It would be informative to indicate the N190 window using shading or similar.

We have added both a 0 point and shading around the N190 region, thank you for these suggestions.

-Fig 2 legend “This data is”  “These data are”

Modification made, thank you.

Reviewer #2: In this paper, the Authors explore how the N190 (ERP component related to body processing) is affected by several, previously unexplored, aspects related to person perception, generally linked to the stimuli’s ecological validity (few/multiple bodies – presence of a real world scene in the background). While in terms of results there are some limitations (due for example to the inconsistency of results related to inversion effects between experiment 1 and 2, and previous literature), the Authors take careful/appropriate conclusions based on their findings. Indeed, natural appearance of the body “in the wild” seem to matter for the N190 amplitude, and for decoding accuracy in general. A side conclusion is also that the N190 cannot be fully accounted for by low-variability within the stimuli set as argued in the past (Thierry et al., 2007).

I think that the manuscript is overall well written, particularly in the intro and discussions, and I enjoyed reading it.

Thank you very much for these comments.

However, I think the paper would benefit with some revisions:

Could several of the effects observed on the N190 be accounted by attentional deployment to body stimuli, rather than some genuine differences due to configural processing? – For example, N190 was smaller (more negative) with multiple people, and more positive with upright stimuli (as opposed to inverted). Both conditions more than one person, and the “weirdness” of an inverted body, supposedly increase the likelihood that someone would gaze to a body/person within a scene, therefore influencing the N190. Generally, it would be useful if in the results the Authors were also consistent using “more negative” or “more positive” – see also points below.

First, we apologize for the lack of consistency with regard to our descriptions of the differences in peak amplitude. We have edited the text to correct this, which we hope helps clarify our results. Second, the reviewer’s observation about attentional deployment is indeed an interesting possibility. We aren’t able to comment on this directly as our design did not include any manipulation or measurement of spatial attention, but we have edited the text to highlight this idea. 

Is that the case that the Authors expected an interaction with the effect of one – to – many people modulating the N190 as a function of inversion? Seems so from what stated in the abstract. If that is the case, could the Author explain why such interaction might be expected?

We did suspect that this might be the case, which we based on the possibility that disruptions of configural processing might be especially evident when only a single body is being viewed free of clutter. We have included some discussion of this point in the revised introduction to motivate our hypothesis more clearly. 

Hemisphere by orientation interactions (both in experiment 1 and 2) – I found these interactions quite intriguing because if inversion affects configural processing, then inverted stimuli body part-based processing should be increasing. Single body part processing seems to rather involve more the left hemisphere (Urgesi et al., 2007; Bracci et al., 2012). Here you instead stronger inversion effects on the right, perhaps reinforcing the interpretation that in the case of the N190 the inversion effect is rather explained by domain general attentional effects rather than configural processing as such.

Thank you for these comments. We had not considered the hemispheric difference in terms of single body parts and we agree that this provides an interesting context for these interactions. 

I found the results of the classification analyses a bit hard to read. Is quite clear what you did – but less clearly written the conclusions you take from them. For example, in the abstract you mention that orientation decoding is also contributed later (after 300ms). However, this is not clearly shown in the result sections. I appreciate the figure 4a and 4b such that all the electrodes can be seen, however, it would be important perhaps also to show a line graph with decoding accuracy on the y axis and time in the x axis across all the electrodes (and/or only taking the electrodes of interest, e.g. those of the N190 or all the posterior electrodes?). In this way would be clearer the “when” of orientation/numerosity/appearance classification reaches significance against chance. Also – do you think would be useful to look at the decoding accuracy of numerosity (exp 1) and appearance (experiment 2), as presented in the figure 4a and 10a without averaging across upright and inverted, and only looking at the upright condition? Of course orientation was task relevant and did not show interactions, but this may be still interesting to look at?

Thank you for suggesting the changes which would clarify some of the decoding results presentation. We have included figures which average the feature ranking across electrodes to provide a visualization of feature importance across timepoints, similar to what you described. We also apologize about the typo in the figure 10a capture (which you have noted in your minor comments) - the caption has been changed to reflect that we only compared upright conditions in the decoding analyses for appearance and numerosity.

The paper would really benefit with some more detailed data visualization, for example it can be quite confusing, particularly for a non-eeg expert reader, to deal with expressions like “more positive”/”less negative” amplitude in one or the other condition. – see minor points specific on the figures.

Again, our apologies for the lack of consistency - we hope our revisions of the text have made this more clear. 

Minor:

E1 Stimuli:

Nice database, what is this database usually used for? Computer vision?

Yes, this is frequently used for pedestrian detection algorithm benchmarking.

In the stimuli example in Figure 1 the face was always non- or barely- visible – is that something that you controlled for, or in some cases the face was fully visible? Would be useful to specify this since, as the Authors acknowledge, that might matter for the inversion effects.

We did not control the visibility of the face and acknowledge this in the revised text as suggested here. 

Figure 2 and Figure 7 do not contain any legend for upright and inverted, also is not entirely clear what the y axis represent, because the labels are absent. I am not entirely sure also what was the purpose of showing these figures?

 We apologize for this and have included these labels in the revised text. We hope that this motivates the inclusion of these figures more clearly. 

Figure 3 – Would be nice that the figure would be a line plot showing the actual component also, would make the results sections clearer – instead of trying to read through “more negative, more positive amplitude”. Also in the captions it says “across conditions” but it should rather be “in each condition”.

We have edited these figures along these lines.

Page 16 – results: Indicate also the contrast between few and many even if it didn’t reach significance; In this section would be also useful to have descriptives related to the differences (Mean values and 95% CIs so that we could make a sense of the extent of the effects.

We have done so in the revised text.

Figure 4a. is this collapsed across upright and inverted conditions or represent only the upright condition? Seems that there is a typo in the caption.

Both this figure caption and the figure 10a caption have been updated to reflect that they represent only the upright condition comparisons.

Discussion Exp 1 and 2: Would be interesting to know whether the inconsistencies between studies on inversion effects affecting the N190 relate to the task-relevance this factor had in the experiments?

We have included some discussion of this point in the revised text.

Results Experiment 2: N190 amplitude: Please include the relevant contrasts at the end when you break down the two significant interactions.

We have done so in the revised text. 

Once again, we would like to thank both reviewers for their comments on our initial submission. We hope that our responses and the changes we have made to the text have adequately addressed the points that they have raised.

---

## [Decision Letter · Decision Letter 1]

6 Mar 2023

PONE-D-22-31727R1Putting People in Context: ERP Responses to Bodies in Natural ScenesPLOS ONE

Dear Dr. Balas,

Thank you for submitting your manuscript to PLOS ONE. After careful consideration, we feel that it has merit but does not fully meet PLOS ONE’s publication criteria as it currently stands. Therefore, we invite you to submit a revised version of the manuscript that addresses the points raised during the review process. Both reviewers are satisfied with your revisions and have only a few minor remaining comments. In particular, both reviewers suggest including example stimulus figures. Once you address all remaning comments I will assess the revisions myself, likely with no need for further review. 

We look forward to receiving your revised manuscript.

Kind regards,

Guido Maiello

Academic Editor

PLOS ONE

Journal Requirements:

Reviewers' comments:

Reviewer's Responses to Questions

**Comments to the Author**

1. If the authors have adequately addressed your comments raised in a previous round of review and you feel that this manuscript is now acceptable for publication, you may indicate that here to bypass the “Comments to the Author” section, enter your conflict of interest statement in the “Confidential to Editor” section, and submit your "Accept" recommendation.

Reviewer #1: (No Response)

Reviewer #2: All comments have been addressed

2. Is the manuscript technically sound, and do the data support the conclusions?

Reviewer #1: Yes

Reviewer #2: Yes

3. Has the statistical analysis been performed appropriately and rigorously? 

Reviewer #1: Yes

Reviewer #2: Yes

4. Have the authors made all data underlying the findings in their manuscript fully available?

Reviewer #1: Yes

Reviewer #2: Yes

5. Is the manuscript presented in an intelligible fashion and written in standard English?

Reviewer #1: Yes

Reviewer #2: Yes

6. Review Comments to the Author

Reviewer #1: The authors have addressed most of my concerns. However, it seems like the example stimuli figures have been removed in this revision. If at all possible, I would strongly encourage the authors to bring these back in (if the experimental stimuli are not allowed to be reprinted, the authors could create new stimuli that are representative of the experimental stimuli). Seeing the example stimuli (both upright and inverted) for both experiments really helps to understand the manipulation in a single glance.

Reviewer #2: I thank the Authors for better characterising the research question specifying that inversion served as a control for body processing specificity. Generally, I think the paper has improved with the revisions and the conclusions taken are supported by the data.

Minor:

Figure 3a 3b - 10a 10b

Please add labels at the axis (electrode n. and timepoint).

Experiment 2: N190 peak latency analysis - You report a marginal effect, please clarify whether this goes in the same direction of exp. 1 - Seems that it is the case from what I read in the discussions.

I would include inverted stimuli examples in the figures for completeness.

7. PLOS authors have the option to publish the peer review history of their article (what does this mean?). If published, this will include your full peer review and any attached files.

Reviewer #1: No

Reviewer #2: No

---

## [Author Response · Author response to Decision Letter 1]

7 Mar 2023

Response to Reviewers

We would once again like to thank our reviewers for their comments on the most recent draft of our manuscript. We were pleased to see that they felt we had adequately addressed their concerns and we have made additional changes to the text (specifically the figures) to respond to their most recent suggestions. We reproduce their comments in full below, with our responses in bold.

Review Comments to the Author

Reviewer #1: The authors have addressed most of my concerns. However, it seems like the example stimuli figures have been removed in this revision. If at all possible, I would strongly encourage the authors to bring these back in (if the experimental stimuli are not allowed to be reprinted, the authors could create new stimuli that are representative of the experimental stimuli). Seeing the example stimuli (both upright and inverted) for both experiments really helps to understand the manipulation in a single glance.

Thank you for this suggestion. We have created new images ourselves (that we retain copyright to) that are now included to illustrate the upright and inverted stimulus conditions in both experiments. 

Reviewer #2: I thank the Authors for better characterising the research question specifying that inversion served as a control for body processing specificity. Generally, I think the paper has improved with the revisions and the conclusions taken are supported by the data.

Thank you for these comments.

Minor:

Figure 3a 3b - 10a 10b

Please add labels at the axis (electrode n. and timepoint).

We have added these as requested in the revised figures. 

Experiment 2: N190 peak latency analysis - You report a marginal effect, please clarify whether this goes in the same direction of exp. 1 - Seems that it is the case from what I read in the discussions.

We have made this explicit in the revised text.

I would include inverted stimuli examples in the figures for completeness.

We have done so.

Thank you again to both of our reviewers for their commentary on the revised draft.

---

## [Editor Report · Decision Letter 2]

13 Mar 2023

Putting People in Context: ERP Responses to Bodies in Natural Scenes

PONE-D-22-31727R2

Dear Dr. Balas,

We’re pleased to inform you that your manuscript has been judged scientifically suitable for publication and will be formally accepted for publication once it meets all outstanding technical requirements.

Kind regards,

Guido Maiello

Academic Editor

PLOS ONE
---

## [Editor Report · Acceptance letter]

20 Mar 2023

PONE-D-22-31727R2 

Putting People in Context: ERP Responses to Bodies in Natural Scenes 

Dear Dr. Balas:

I'm pleased to inform you that your manuscript has been deemed suitable for publication in PLOS ONE. Congratulations! Your manuscript is now with our production department. 

Kind regards, 

on behalf of

Dr. Guido Maiello 

Academic Editor

PLOS ONE